# Critical success factors influencing business intelligence adoption: Evidence from Yemen

Amira Taha Al-Adimi [1,2]*, Mokhtar Mohammed Ghilan[1], Walid Shaher Yousef [3], Abdullatif Ghallab [2]

1 Department of Information Systems, Sana'a University, Yemen, 2 Department of Computer Science, University of Science and Technology, Yemen, 3 Department of Information Technology, University of Science and Technology, Yemen

* amiraadimi@gmail.com

## Abstract

Over the past five decades, decision support systems have evolved into business intelligence (BI) systems, which are now a strategic priority for many organizations. However, despite their widespread adoption, many BI projects fail, highlighting the need to identify Critical Success Factors (CSFs). While CSFs are well-studied in developed economies, there is a significant lack of empirical research in developing countries, which face unique challenges. This gap is particularly evident in Yemen, where BI adoption is still in its early stages of adoption. This study addresses this gap by investigating the CSFs for BI adoption in the Yemeni context. To do this, we develop and validate a novel integrated TOEP framework by combining the Technology-Organization-Environment (TOE) framework with the process-oriented Yeoh and Koronios model. Furthermore, we employ the rigorous Rough Stepwise Weight Assessment Ratio Analysis (R-SWARA) method, a multi-criteria decision-making approach adept at handling expert judgment uncertainty, to rank the CSFs. The results reveal that competitive pressure, data quality, clear vision, and change management are the most significant drivers in Yemen. However, in contrast to stable economies, information-sharing culture and system integration are currently the greatest challenges to these systems in the Yemeni context. The findings provide actionable insights for managers and policymakers in similar challenging environments, offering a contextualized model for successful BI adoption.

## Introduction

In an era defined by global market competitiveness, digital transformation has become a strategic imperative for modern enterprises. This shift necessitates the integration of Industry 4.0 technologies, with a particular emphasis on advanced analytics and Business Intelligence (BI) systems. While the conceptual origins of BI date back to 1990, recent trends indicate a significant surge in prioritization among

**Data availability statement:** All relevant data are within the manuscript.

**Funding:** The author(s) received no specific funding for this work.

**Competing interests:** The authors have declared that no competing interests exist.

Chief Information Officers (CIOs), leading to substantial capital investments within their organizations. As BI adoption is projected to permeate all industrial sectors in the coming years, the role of these solutions has transitioned from a discretionary advantage to a critical necessity for organizational survival and strategic positioning in today's dynamic economic landscape [1–3].

Despite the recognized benefits of BI systems, empirical evidence consistently reports a high failure rate of BI initiatives, particularly in developing and unstable environments [4,5]. These failures are rarely attributed to technological limitations alone; instead, they often stem from organizational, environmental, and process-related challenges that hinder effective implementation and utilization [6]. Consequently, identifying and prioritizing the Critical Success Factors (CSFs) that influence BI adoption has become a central concern in both academic research and professional practice [7].

Existing studies on BI adoption and its CSFs are predominantly concentrated in developed and institutionally stable economies, where technological infrastructure, regulatory frameworks, and organizational capabilities are relatively mature [8–10]. While these studies have generated valuable insights, their findings cannot be readily generalized to fragile and conflict-affected contexts characterized by limited resources, weak institutional structures, and volatile operating conditions [11]. This creates a significant research gap, as organizations operating in such environments face fundamentally different challenges that may alter the relative importance of traditional CSFs.

Yemen represents a particularly underexplored context in this regard. In recent years, organizations in Yemen, especially in sectors such as telecommunications, banking, humanitarian operations, and public administration, have begun to adopt BI solutions to enhance transparency, coordination, and decision-making [4,5]. However, BI adoption in Yemen remains at a nascent stage and is confronted by severe challenges, including infrastructure limitations, data fragmentation, organizational instability, and regulatory uncertainty [6,7]. Despite these challenges, there is a notable absence of empirical studies that systematically investigate the drivers and barriers of BI adoption within the Yemeni context.

From a theoretical perspective, technology adoption research has traditionally relied on generalized frameworks such as the TOE framework to explain organizational-level adoption decisions. While TOE provides a robust and flexible structure, it lacks the granularity required to capture the process-oriented dynamics that are critical to BI implementation. Conversely, BI-specific models, most notably the Yeoh and Koronios model, offer valuable insights into BI-related organizational and technological factors but largely overlook the influence of external environmental pressures. As a result, neither framework alone provides a sufficiently comprehensive lens for analyzing BI adoption in fragile and resource-constrained environments.

To address these limitations, this study proposes a novel integrated framework—Technology–Organization–Environment–Process (TOEP)—which synthesizes the strengths of the TOE framework with the BI-specific insights of the Yeoh and Koronios model. The explicit inclusion of the Process dimension represents a key

theoretical advancement, as it captures dynamic implementation mechanisms such as change management, project champion, and balanced team and project methodology that are not adequately represented in existing adoption models. This integration provides a more holistic and context-sensitive framework for analyzing BI adoption, particularly in developing and unstable economies.

In addition to its theoretical contribution, this study introduces methodological advancement by employing the Rough Stepwise Weight Assessment Ratio Analysis (R-SWARA) method to prioritize BI CSFs under conditions of expert judgment uncertainty. Unlike conventional multi-criteria decision-making methods that rely on extensive pairwise comparisons, R-SWARA reduces cognitive burden on experts while maintaining analytical rigor, making it particularly suitable for under-researched contexts where expert availability is limited. Accordingly, the objectives of this study are twofold:

**(RO1)** to develop a comprehensive conceptual framework for BI adoption by integrating technological, organizational, environmental, and process dimensions.

**(RO2)** to empirically prioritize the CSFs influencing BI adoption in the Yemeni context using the R-SWARA method.

By addressing these objectives, this research provides context-specific theoretical insights and practical guidance for managers, policymakers, and system developers operating in fragile environments. Moreover, it contributes to the broader BI adoption literature by demonstrating that the hierarchy of CSFs is not universal but highly contingent upon local institutional and environmental conditions.

## Literature review and background

### Business intelligence

The term "business intelligence" was first introduced by Howard Dresner in 1989, who defined it as a set of concepts and methodologies to improve business decisions using facts and information from supporting systems [12]. Over time, this concept has evolved into a process that captures, analyzes, and transforms a company's raw data into valuable information to enhance decision-making and business operations [13–16]. The core functionalities of BI involve integrating diverse data sources, analyzing large datasets, and providing tailored analytical solutions for knowledge discovery [17]. These functionalities are essential for informed decision-making and gaining a competitive advantage [14,18,19]. In addition, BI helps organizations foster innovation by enhancing their dynamic capabilities, allowing them to adapt to changing market conditions by analyzing the surrounding environment [20]. The strategic benefits of BI also include improved operational efficiency, better risk management, and increased customer satisfaction. Consequently, BI is now an integral pillar for modern organizational success, which amplifies the value of studies that examine the CSFs for adopting these systems. However, this very strategic importance underscores a critical paradox: despite its recognized benefits, a high rate of BI project failure persists. This contradiction suggests that the successful adoption of BI is not a given and is likely influenced by a complex set of factors that extend beyond the technology itself, necessitating a deeper investigation into what truly drives successful implementation.

### TOE framework

The Technology-Organization-Environment (TOE) framework is a popular choice for studying systems adoption because it provides a comprehensive, holistic, and flexible perspective [21]. Unlike many other models that focus on a single aspect, TOE offers a comprehensive view of the human and non-human factors influencing this process [22]. The primary strength of the TOE framework is its ability to integrate multiple levels of influence. It recognizes that a technology adoption decision is not made in a vacuum but is shaped by a complex interplay of internal and external factors [21]. The TOE framework considers three distinct contexts that influence a firm's decision to adopt and implement a new technology [22]. The three core contexts are the technological context, organizational context, and environmental context, which can significantly influence the successful adoption and effective utilization of new technologies [14].

The TOE framework's general nature makes it highly adaptable to a wide range of technologies and industries. Researchers have successfully used it to study the adoption of everything from e-commerce systems and enterprise resource planning (ERP) software to cloud computing and social media in diverse sectors, including manufacturing, retail, and telecommunications. This broad applicability solidifies its status as a robust and reliable theoretical model for organizational-level technology adoption. Therefore, it was adopted as one of the foundations for the study's framework.

However, the very generality that makes the TOE framework widely applicable also limits its utility for a complex, context-specific system like BI. While it provides a strong structural guide, it inherently lacks the specificity to capture the unique, process-oriented success factors essential for BI implementation [23].

### Yeoh and Koronios model

The work of Yeoh and Koronios is a foundational framework for studying BI adoption because it was one of the first to specifically identify and categorize CSFs tailored to the unique complexities of BI systems [24]. Unlike earlier studies that often treated BI as just another IT project, Yeoh and Koronios recognized that BI has distinct characteristics. Its complex architecture, reliance on data warehouses, and close link to business strategy required a new model. Their research was among the first to bridge this gap between general IT adoption theories and the specific needs of BI implementation.

The framework provides a specific understanding of the factors influencing BI success. It organizes CSFs into clear categories, including organizational factors, technological factors, and process-related factors [25]. This business-centric approach sets it apart. It emphasizes that a BI project's success is not just about the technology; it's heavily dependent on how the organization leverages and manages the process and strategic factors of the project. Its key innovation was the explicit inclusion of the process dimension, highlighting that success depends heavily on structured workflows, project management, and change management [23,26]. This model successfully bridges the gap between general theory and the specific implementation needs, offering a tailored set of CSFs. Subsequent research has frequently used the Yeoh and Koronios model as a benchmark to further explore and validate new factors in different contexts [24].

However, this specific model suffers from a major theoretical shortfall: it largely overlooks the broader environmental context. Crucial external factors, such as competitive pressure and government regulations (which are central to the framework), are not adequately addressed. This omission constitutes a significant limitation in today's globalized and highly regulated business environment.

### Critical success factors

CSFs are the essential areas in which an organization's performance must be satisfactory to ensure its competitive success [27,28]. They demand continuous managerial attention and are commonly used to guide the execution of various strategies and programs [29]. As such, focusing on these factors is crucial for achieving organizational success [30]. Studying the CSFs of BI adoption is important because it provides a clear roadmap for organizations to follow, increasing their chances of success and avoiding common pitfalls. However, despite the acknowledged importance of CSFs for organizational success, there is limited research on the CSFs of successful adoption of BI [5]. Therefore, a thorough analysis of the dimensions and CSFs is essential for the effective implementation of BI systems.

Identifying CSFs is only the first step; understanding their relative importance is crucial for effective resource allocation. Multiple Criteria Decision Making (MCDM) methods are designed for this purpose, with studies frequently employing techniques like the Analytical Hierarchy Process (AHP) and Fuzzy (FAHP) to rank CSFs [31,32].

### Multi-criteria decision making in CSF identification

MCDM is a field within operational research. It focuses on identifying optimal solutions in complex situations that involve numerous, often competing, criteria and goals [33]. Several scholars have utilized MCDM to examine the BI adoption

system. For instance, Halim et al. [31] evaluated the CSFs for implementing a Data Warehouse and Business Intelligence (DW/BI) system at the Indonesian financial institutions. Using the Analytical Hierarchy Process (AHP) method, the study determined the most influential CSFs across three dimensions: people, process, and technology. Overall, the people dimension was found to be the most influential, while the source system was identified as the most impactful sub-criterion. Furthermore, Alabaddi et al. [32] also worked on identifying the most significant CSFs for BI using MCDM methods. The study used the Fuzzy Analytical Hierarchy Process (FAHP) with 12 experts from six countries (Egypt, United Arab Emirates, Saudi Arabia, China, Hong Kong, and Brazil). It found that organizational and technological factors are top priorities, specifically top management support, proper vendor selection, and adequate resources. The results are contradicted and may not be valid for specific countries' unique market conditions. Therefore, the authors call for further investigation.

A critical analysis of MCDM methods revealed that many researchers have used AHP and FAHP as MCDM methods. These methods are based on pairwise comparison, which can be challenging for respondents due to the large number of required pairwise comparisons. This often leads to respondent fatigue and potentially distorted answers. This cognitive burden is a critical drawback in contexts like ours, where BI experts are scarce; therefore, the quality of data can be easily compromised by lengthy surveys. To mitigate this, our study employed R-SWARA, one of the most effective and modern Multi-Criteria Decision-Making (MCDM) methods. R-SWARA streamlines the data collection process by requiring respondents to simply rank criteria according to their importance based on respondents' beliefs, facilitating more accurate and reliable responses [34]. This makes it particularly suitable for a nascent research environment like Yemen, where securing deep expert engagement is challenging.

The preceding analysis reveals that the comprehensive but generic TOE framework fails to capture the process-specific nuances of BI adoption, while the specific but internally focused Yeoh and Koronios model neglects vital external environmental pressures. Therefore, our study proposes the TOEP framework by integrating these two models. This integrated framework leverages the strengths of both: the TOE's comprehensive external and internal scope and the Yeoh and Koronios model's specificity on BI processes. It provides a complete and more nuanced lens to investigate BI adoption, particularly in a complex environment like Yemen, where both external pressures (environment) and internal implementation (process) are critical.

In Yemen, the adoption of BI systems is in its early stages but is already beginning to reshape competitive practices, especially in data-intensive sectors like telecommunications, banking, and commerce. Academically, this nascent phase offers a critical opportunity to study the factors that drive successful BI adoption. The novel integrated framework and a robust, context-appropriate methodology allow us to effectively investigate the CSFs for BI adoption in the under-researched and challenging context of Yemen, where BI is beginning to reshape competitive practices in key sectors.

## Theoretical framework

A key strength of research is the choice of theories used to build its model. The TOE framework is highly effective for studying technology adoption because it captures the complicated nature of technology adoption within organizations, providing a holistic, organizational-level perspective [35–38]. A key strength is its adaptability, allowing it to be integrated with other models like the Diffusion of Innovations (DOI) and Institutional Theory to explore various contexts [39]. Furthermore, the TOE's combination with the Technology Acceptance Model (TAM) has proven effective in investigating the adoption of Industry 4.0 technologies [40]. Moreover, its integration with the Diffusion of Innovation (DOI) theory has enhanced studies on e-business adoption [41]. This adaptability highlights the TOE framework's utility as a flexible theoretical tool for examining technology innovation across diverse settings.

Furthermore, Yeoh and Koronios' model is a model specifically designed for BI adoption, focusing on technology, organization, and process dimensions to identify CSFs and maximize return on investment (ROI) [23]. The model's key innovation is its emphasis on the process dimension, which is considered essential for successful implementation. It highlights the importance of structured workflows, effective project management, and strategic change management, including

training and communication, to ensure efficient project execution, minimize costs, facilitate organizational transition, and support continuous improvement [24].

A theoretical model was developed to investigate the adoption of BI systems. This model synthesizes the TOE framework with the Yeoh and Koronios model. The combined framework provides a broader, more general perspective while also focusing on BI-specific aspects. The resulting model encompasses four dimensions: technology, organization, environment, and process [23,24]. Synthesizing these two frameworks is shown in Fig 1.

The integration of the 'Process' dimension as a standalone pillar in the proposed TOEP framework is fundamentally grounded in the specialized nature of Business Intelligence (BI) systems, which differ significantly from traditional IT implementations. According to Yeoh and Koronios (2010), conventional adoption models like the TOE framework often focus on static factors such as organizational size or technical readiness, thereby overlooking the procedural dynamics that determine BI success [23]. They argued that because BI is a business-driven, evolutionary initiative rather than a one-time software installation, it requires a dedicated focus on the implementation methodology to ensure continuous alignment with strategic goals [42]. This procedural independence is further justified by the need for an incremental delivery approach, which allows for iterative development to maintain ongoing management patronage and mitigate risks associated with large-scale data integration [23]. Furthermore, since BI implementation triggers a profound shift in an organization's decision-making culture, managing this transition requires specific "procedural drivers," namely, a high-level project champion and a balanced team to navigate cross-functional complexities [43]. By isolating these factors from general organizational traits, the TOEP framework provides the necessary theoretical granularity to analyze how structured management processes can overcome the unique operational barriers found in emerging and unstable environments like Yemen.

The proposed TOEP framework does not merely combine two models; rather, it functions as a theoretical bridge that reconciles the macro-level organizational focus of TOE with the micro-level procedural rigor of Yeoh and Koronios. This integration is particularly novel in its application to a conflict-affected, emerging economy, where the high stakes of BI implementation demand a framework that simultaneously accounts for external environmental volatility and internal process-driven resilience.

The proposed TOEP framework defines four critical dimensions and their sixteen respective factors essential for successful adoption, as illustrated in Fig 2:

**Technology factors:** Data quality, relative advantages, compatibility, complexity, system integration, IT infrastructure, and a business-driven, scalable, and flexible technical framework.

**Organization factors:** Top management support, adequate resources, clear vision and business alignment, organization size, and an information-sharing culture.

**Environmental factors:** Competitive pressure and regulations.

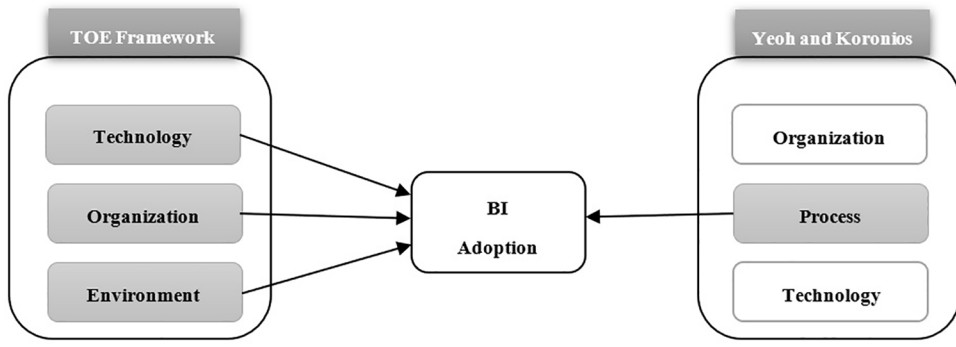

**Fig 1. Theories mapping.**

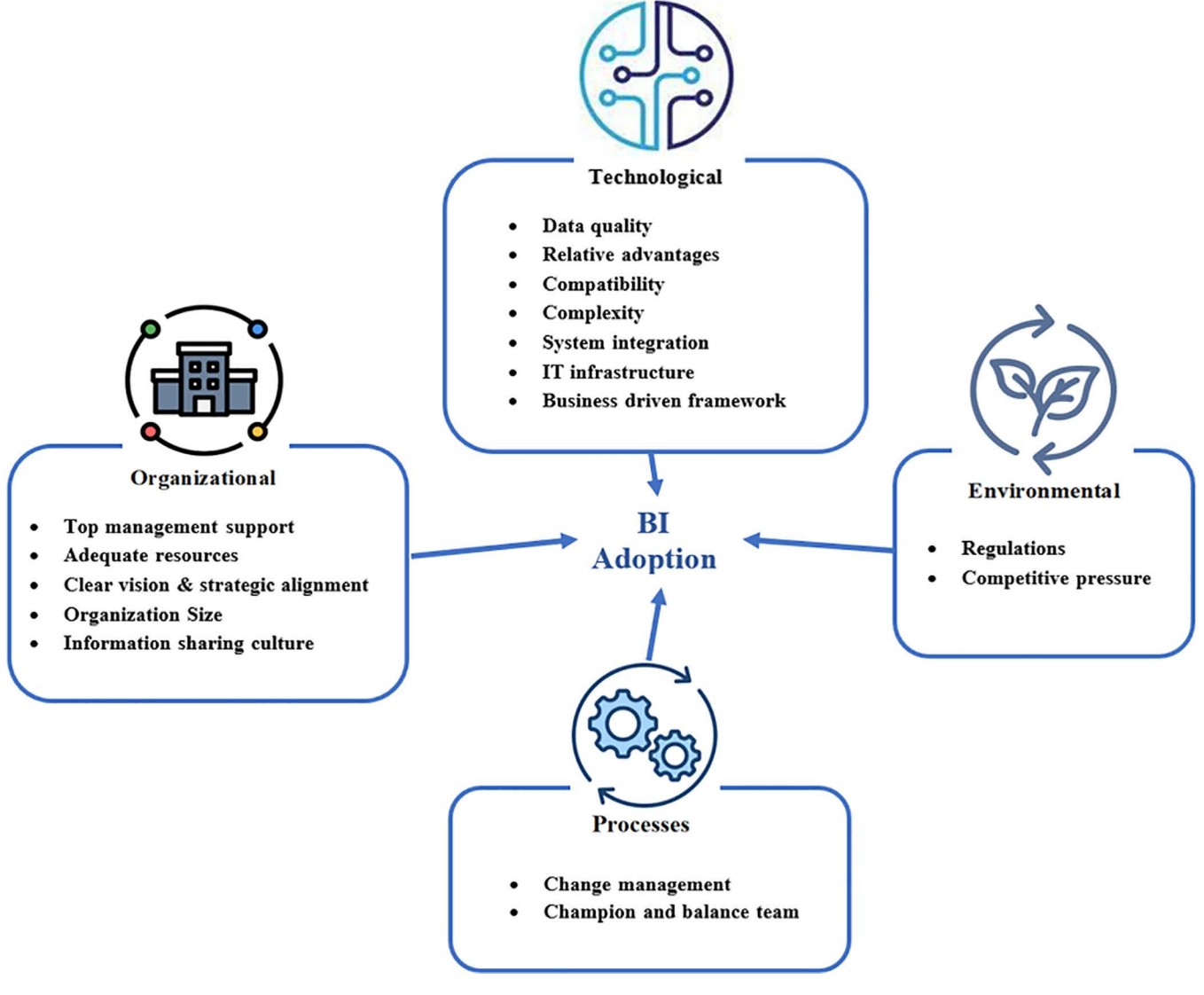

**Fig 2. The conceptual model.**

**Process factors:** A balanced team led by a champion and effective change management.

This study will use these critical factors to investigate technology adoption specifically within large and medium-sized Yemeni organizations. The next section explains the CSFs of the proposed framework.

## Technological context

Understanding the adoption of BI requires an examination of critical technological factors. The technological context includes an organization's internal and external technologies, procedures, and resources that support and drive innovative activities [44]. This study investigates seven key technological factors influencing BI adoption within Yemeni organizations: data quality, relative advantage, compatibility, complexity, system integration, IT infrastructure, and a business-driven, scalable & flexible technical framework.

## Data quality

Data quality is broadly defined by its adherence to criteria such as accuracy, comprehensiveness, consistency, and completeness, which collectively ensure the reliability and trustworthiness of the data within the data warehouse [45,46]. Specifically, consistency and completeness are vital, as they significantly influence the effectiveness and accuracy of all subsequent data analytics processes [47]. The reliability, relevance, accuracy, and predictability of data form the critical foundation necessary for the successful adoption and effective utilization of BI systems [48]. Consequently, prioritizing high data quality is essential not only for achieving valuable insights but also for gaining a competitive advantage [49]. Therefore, understanding and ensuring data accuracy is crucial for any organization that aims to effectively leverage its data assets. Thus, this study proposes that data quality significantly affects BI adoption.

## Relative advantage

Relative advantage is defined as the degree to which an innovation is perceived as superior to existing alternatives [50]. Compared to traditional systems, BI offers a compelling array of advantages, positioning it as a strategically advantageous option for organizations [51]. This perceived relative advantage of BI is influenced by a range of factors, including the organization's size, its specific resources, unique strategic goals, and the availability of skilled personnel to effectively manage and utilize BI systems [52]. The potential advantages often serve as the primary motivation for BI adoption, though their impact is context dependent. Therefore, our study supposes that relative advantage affects BI adoption.

## Compatibility

According to Rogers (1995), compatibility is the degree to which an innovation aligns with a social system's existing values, beliefs, and practices [53], as well as the experiences and needs of potential users [40]. For successful BI adoption, a system's compatibility with current organizational processes is crucial [54]. If a BI system is incompatible with existing systems, it may not fully utilize their features and functionalities, which reduces its overall value to the organization [55]. Therefore, this study supposes that compatibility affects BI adoption.

## Complexity

Rogers (2003) defines perceived technological complexity as the degree to which an innovation is seen as difficult to understand and use, a crucial factor influencing technology [53]. This is especially relevant for BI systems, as data warehousing projects have a unique level of complexity that sets them apart from traditional system development [56]. High complexity can significantly impede user adoption by creating usability challenges and hindering a user's ability to effectively navigate the system [57]. In contrast, a simple and intuitive BI system with minimal training requirements can facilitate rapid adoption and more effective utilization of its functionalities [14]. Therefore, this study supposes that complexity affects BI adoption.

## System integration

System integration refers to a system's ability to seamlessly interact and communicate with other existing systems and databases within an organization [58]. This capability is critical for BI systems, given their comprehensive data-driven nature. Effective BI requires robust data integration to collect and consolidate information from diverse sources, enabling comprehensive analysis and valuable insights. However, this process presents significant challenges [59]. Difficulties in integration can lead to project delays and hinder a BI system's effective implementation and utilization [60]. Moreover, if a BI system cannot integrate effectively with legacy systems, it may not fully leverage their features, which limits its functionality and diminishes its value to the organization [55]. Therefore, our study proposes that successful system integration is a crucial factor in the successful adoption of BI.

## IT infrastructure

IT infrastructure is the foundational collection of hardware, software, networking, and other foundational components that support an organization's business applications [61]. A flexible and scalable infrastructure is crucial for accommodating evolving information needs and changing business requirements [31]. For BI adoption, this infrastructure must be robust enough to handle large volumes of data, perform complex analytics, and ensure seamless data integration across various systems and sources [35]. Therefore, the proposed model suggests that IT infrastructure affects BI adoption.

## Business driven, scalable & flexible technical framework

For a BI system to be truly effective, it must be designed as a business-driven, scalable, and flexible technical framework [23]. This design approach ensures the system aligns with an organization's strategic vision and business needs [23]. A key motivation for adopting a BI system is its potential to help an organization achieve strategic objectives and overcome challenges. This requires a system that is not only robust but also flexible and scalable, capable of evolving with changing business requirements and enabling data-driven decision-making [62]. A well-designed BI system provides a strong foundation of data sources and analytical features that can be adapted to current and future needs, making it a sustainable, long-term solution [23]. Consequently, this factor is selected to be one of the CSFs of our proposed framework.

## Organizational context

To fully comprehend the effective utilization of BI within an organizational context, it is crucial to consider the organizational critical factors [63]. This study proposes five organizational CSFs: top management support, adequate resources, clear vision and business alignment, organization size, and information sharing culture for BI adoption.

## Top management support

Top management support is critical for any project's success. This involves senior leadership's active commitment and involvement, including the allocation of necessary resources, delegation of authority, and risk management. Securing this support is considered a crucial challenge in adopting any new technology, including BI [64]. It requires senior managers to recognize the importance of the technology and be convinced to invest in it. Without adequate top management support, a BI project's full potential may not be realized [65]. Thus, this study suggests that top management support affects BI adoption.

## Adequate resources

Adequate organizational resources, encompassing sufficient financial, technical, and human capital, are crucial for the successful adoption of new technologies [66]. Organizations with greater access to these resources typically demonstrate enhanced adaptability and agility when integrating new systems [67]. Consequently, the availability of such resources significantly improves the likelihood of successful BI system adoption and effective utilization [14,23]. Therefore, our framework supposes that adequate resources affect BI adoption.

## Clear vision and business alignment

Given the strategic nature of BI systems, a clear and well-defined business vision is essential to guide their successful adoption [23]. The success of BI systems pivots a clear alignment between the BI strategy and the main business strategy. This ensures that BI initiatives directly support and contribute to key organizational objectives, ultimately driving a competitive advantage. Because BI systems are strategic assets, a clear vision is critical for successful implementation. An ambiguous business vision can negatively impact the outcomes of BI adoption [23]. Thus, our framework suggests that a clear vision affects BI adoption.

## Organization size

Larger organizations are better equipped to absorb the costs and risks associated with technology adoption [68]. Accordingly, the size of an organization plays a critical role in its ability to effectively adopt and integrate new technologies [53]. Organizational size can be measured by various factors, including the number of employees (staff size) and the organization's overall budget [53]. Due to their inherent complexity, BI systems necessitate significant investments in terms of infrastructure, expertise, data warehousing, system integration, and data sharing. The resource-intensive nature of BI implementation presents a greater challenge for smaller organizations compared to larger enterprises. Therefore, this study supposes that organization size affects BI adoption.

## Information sharing culture

An organization's information sharing culture refers to the shared values and practices governing its acquisition, management, and use of information. As BI systems heavily rely on data analytics, the seamless and secure flow of information across organizational boundaries and with external partners is a critical challenge. Organizations with a culture that promotes data accessibility, utilization, and sharing are better equipped to leverage BI systems to their full potential, leading to improved decision-making and a competitive advantage [14]. However, this is not without challenges. A transparent exchange of information with partners, while beneficial for transactions, may sometimes weaken negotiation capabilities. Furthermore, the need to maintain data privacy and confidentiality can hinder the free flow of information crucial for effective BI systems. This is especially challenging for a centralized data warehouse, which must serve as the primary source for all organizational data while upholding security protocols. Therefore, our proposed framework supposes that information sharing culture affects BI adoption.

## Environmental context

This study examines the influence of external forces on an organization's technology adoption. It specifically investigates how competitive pressure and regulations key factors within the environmental context impact the adoption of BI.

## Competitive pressure

"Competitive advantage" refers to the level at which a technology provides a competitive edge [53]. Competitive pressure serves as a significant driver for the adoption of innovative technologies within organizations [69]. In today's rapidly evolving business environment, organizations are under constant and increasing pressure to improve their performance and maintain a competitive advantage. Organizations facing significant competitive pressure and striving for continuous performance improvement are more likely to adopt BI systems to leverage data-driven insights and gain a competitive edge. Therefore, competitive pressure serves as a significant driver for the adoption of BI systems within organizations [70]. Thus, this study suggests that competitive pressure affects BI adoption.

## Regulation

The regulatory environment consists of government laws, regulations, and policies that significantly impact the adoption and diffusion of new technologies [71]. The effect of these regulations on technological innovation is varied; they can either foster innovation by creating a supportive framework or hinder it if overly stringent [72]. For BI systems, specific challenges arise from regulations concerning data privacy, security, and data sharing compliance [73]. A robust legal and regulatory framework is therefore essential to facilitate the secure and ethical flow of information, which is critical for the successful adoption and effective utilization of BI. Therefore, the proposed framework suggests that regulation affects BI adoption.

## Processes context

The process dimension is a crucial factor in the adoption of BI systems [23]. This study aims to understand its influence by focusing on two key elements: the champion & balance team and change management.

### Champion & balance team

A champion with strong business knowledge is crucial for the successful adoption of BI, as they can anticipate and overcome obstacles [23]. This individual's role is to facilitate collaboration and bridge the gap between business units and the BI team, ensuring that data requirements are met and that functional barriers are removed. To support the champion, an organization must assemble a high-performing project team. This team requires a diverse set of skills, including technical expertise, effective communication, and strong project management abilities [74]. Given the inherent complexity of BI systems, the involvement of these highly skilled and experienced professionals is necessary for successful adoption. Thus, the proposed framework supposes that the existence of a champion and a balanced team for a BI project affects such adoption.

### Change management

Change management is a continuous process of organizational transformation, enabling an organization to adapt its direction, structure, and capabilities to meet evolving stakeholder needs [75]. The successful adoption of BI systems inevitably leads to significant organizational changes. Therefore, effective change management is crucial to mitigate potential disruptions, ensure a smooth transition, and maximize the benefits of BI adoption. This involves a multi-faceted approach, including user awareness programs, comprehensive training, and transparent communication channels to ensure the successful integration of the new technology. Therefore, our framework proposes that change management affects BI adoption.

## Research methodology

This study adopts a structured two-phase research methodology adapted from the frameworks established by Mackenzie and House [76] and McGrath [77], as depicted in Fig 3. The initial exploratory phase comprised a systematic and comprehensive literature review, which facilitated the construction of the research model. This phase culminated in the finalization of sixteen CSFs for BI implementation, as detailed in Table 1.

This framework was specifically built by integrating the TOE framework with the model proposed by Yeoh and Koronios. Next, the confirmatory phase will use the R-SWARA method to empirically validate the proposed framework and determine the relative importance of its CSFs using newly collected data.

### Rough-SWARA method

The Rough Step-Wise Weight Assessment Ratio Analysis (R-SWARA) is a robust multi-criteria decision-making (MCDM) technique introduced by Zavadskas et al. (2018) to evaluate the relative importance of criteria [92]. By integrating the traditional SWARA method with rough set theory, this approach is specifically engineered to mitigate the subjectivity and uncertainty inherent in human judgment.

R-SWARA offers several distinct advantages over other prominent MCDM techniques, such as the Analytic Hierarchy Process (AHP), the Analytic Network Process (ANP), and the Best-Worst Method (BWM). It provides a more streamlined and efficient framework, requiring fewer pairwise comparisons, which enhances both transparency and computational clarity [93,94]. The use of "rough numbers" is particularly valuable as it mitigates the inherent subjectivity associated with human judgment, making the weight determination process highly reliable [92]. Moreover, rough number-based models are inherently designed to accommodate diverse expert perspectives and diminish the impact of outliers by defining lower and upper approximations, thus managing vagueness without strict reliance on traditional consensus measures.

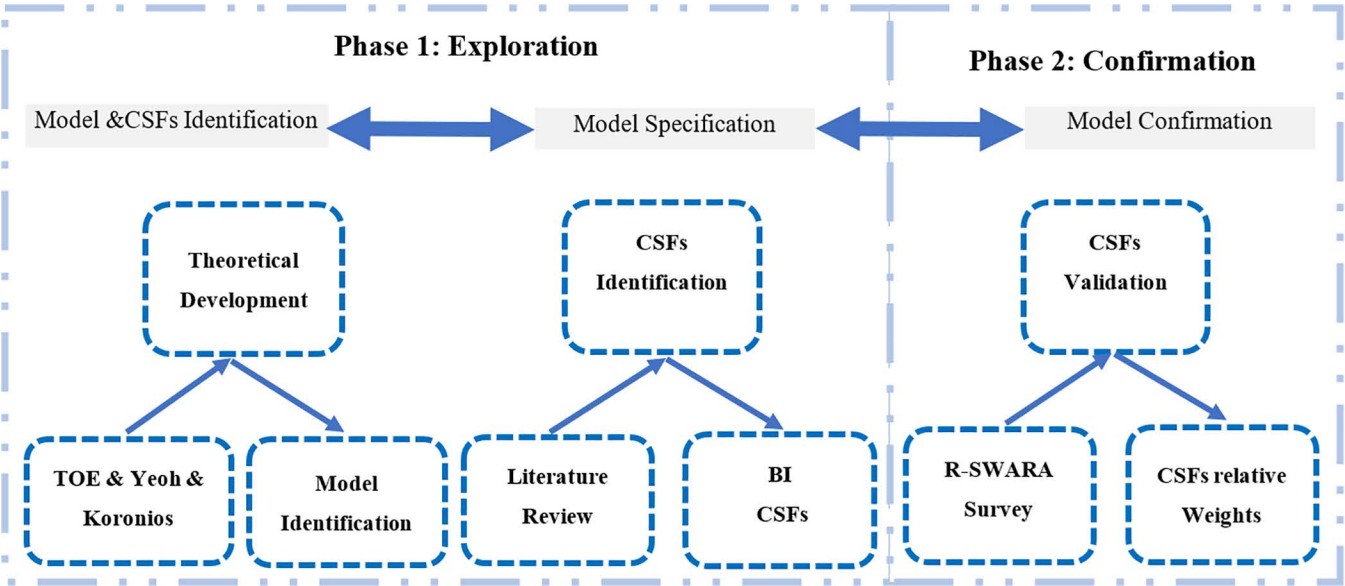

**Fig 3. Research design.**

**Table 1. Summarizes the studies of CSFs for BI adoption from literature.**

| Dimension | CSFs | References |
|---|---|---|
| Technological | Information/data quality | [47,78] |
| | Relative advantages | [14,79] |
| | Compatibility | [54,80] |
| | Complexity | [56,81] |
| | Business driven, scalable & flexible technical framework | [23,82] |
| | System integration | [59,81] |
| | IT infrastructure | [83,84] |
| Organizational | Top management support | [23,59] |
| | Adequate resources | [23,85] |
| | Clear vision and business strategic alignment | [23,81] |
| | Organization size | [53,86] |
| | information sharing culture | [14,87] |
| Environmental | Competitive pressure | [70,88] |
| | Regulations | [89,90] |
| Processes | Champion and balance team | [23,91] |
| | Change management | [23,81] |

Since its inception, the R-SWARA method has gained significant traction within the research community and has been successfully applied across diverse fields to address complex decision-making challenges [95].

In this study, R-SWARA was employed as a systematic tool to prioritize the Critical Success Factors (CSFs) within the proposed framework. To provide a clear and accessible overview of the research stages, Fig 4 presents a graphic summary of the methodological workflow, illustrating the transition from expert judgment to the final weight derivation. The mathematical procedure for the R-SWARA method involves the following systematic steps:

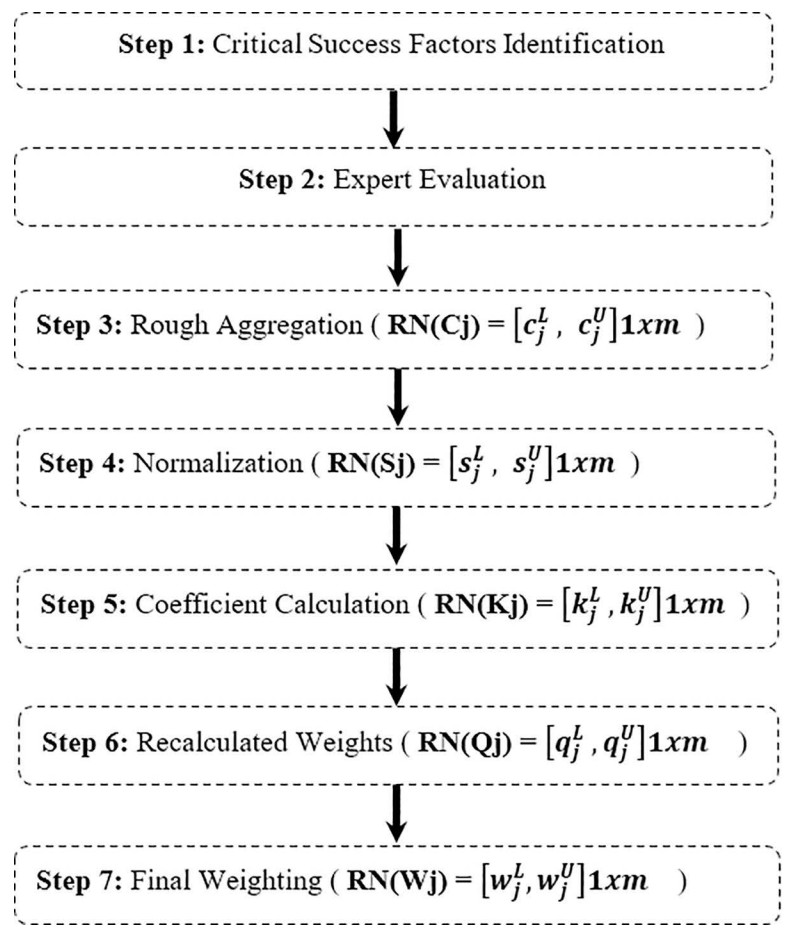

**Fig 4. R-SWARA steps Flow.**

**Step 1:** Define the set of CSFs that will be used in the decision-making process.

**Step 2:** Initiate a group of K experts to evaluate the importance of CSFs. A crucial initial step involves ranking the criteria in descending order of perceived importance. Subsequent pairwise comparisons are then conducted, beginning with the second criterion, c2, to determine its relative importance compared to the first criterion, c1. This iterative process continues for all subsequent criteria cn, enabling the establishment of a clear hierarchy of importance.

**Step 3:** Aggregate individual expert responses (K1, K2,..., Kn) into a collective rough matrix Cj using equations (1–6) mentioned by Zavadskas et al. [92].

$$RN(C_j) \;=\; \left[ c_j^L, \; c_j^U \right] \mathbf{1xm}$$

(1)

Where $j$ = alternative to $j$ criterion; $m$ = criterion under consideration; $c_j^L$ = the lower limit of rough number; $c_j^U$ = the upper limit of number, which indicates the extent to which the best criterion $c_j$ is more significant than other criterion $c_{j-1}$

**Step 4:** Normalization of the matrix RN (Cj) to obtain the matrix RN (S j) as shown in equation (2):

$$RN(S_j) \;=\; \left[ s_j^L, \; s_j^U \right] \mathbf{1xm}$$

(2)

Where $s_j^L$ = the lower limit of rough number; $s_j^U$ = the upper limit of rough, which indicates the extent to which the best criterion $c_j$ is more significant than the criterion $c_{j-1}$.

The matrix elements RN(Sj) are obtained by applying equation (3)

$$RN(Sj) = \frac{\left[ c_j^L, \ c_j^U \right]}{\max\limits_r \left[ c_r^L, c_r^L \right]}$$

(3)

Where $c_j^L$ = the lower limit of rough number; $c_j^U$ = the upper limit of rough number.

The first element of matrix RN (Sj) is respectively $\left[ s_j^L, \ s_j^U \right] = [1.00, \ 1.00]$ since j = 1. The residual element j > 1 in equation (9) can be calculated using equation (4).

$$RN(Sj_{1=2}^m) = \left[ \frac{c_j^L}{\max(c_r^L)}; \ \frac{c_j^U}{\max(c_r^U)} \right] \ 1xm$$

(4)

Where $c_j^L$ = the lower limit of rough number; $c_j^U$ = the upper limit of rough number; max $c_j^L$ the maximum value of the lower limit of rough number; max $c_j^U$ = the maximum value of t he upper limit of rough number.

**Step 5:** calculate the matrix RN(Kj) using equation (5):

$$RN(K_j) = \left[ k_j^L, \ k_j^U \right] 1xm$$

(5)

By applying equation (6):

$$RN(K_j^m) = \left[ s_j^L + 1, s_j^U + 1 \right] 1xm \quad j = 2, 3, \ldots, m$$

(6)

Where j = 2,3,…m; m = criterion under consideration; $k_j^L$ = the lower of the coefficient; $k_j^U$ = the upper limit of the coefficient; $s_j^L$ = the lower limit of rough number; $s_j^U$ = the upper limit of rough number.

Step 6: Determine the matrix of recalculated weights RN(Qj) (7):

$$RN(Q_j) = \left[ q_j^L, \ q_j^U \right] 1xm$$

(7)

Where $q_j^L$ = the lower limit of the recalculated weight; $q_j^U$ = the upper limit of the recalculated weight; m = criterion under consideration.

The elements of the matrix in equation (13) are obtained using equation (8):

$$RN(Q_j) \left[ q_j^L = \begin{cases} 1.00 & j = 1 \\ \frac{q_{j-1}^L}{k_j^U} & j > 1 \end{cases}, \ q_j^U = \begin{cases} 1.00 & j = 1 \\ \frac{q_{j-1}^U}{k_j^L} & j = 1 \end{cases} \right]$$

(8)

Where j-1 indicates the previous criterion in relation to j.

**Step 7:** Calculate the matrix of relative weight values RN (Wj) (9):

$$RN(Wj) = \left[ w_j^L, \ w_j^U \right] 1xm$$

(9)

Where $w_j^L$ = the lower limit of the criteria weight of the rough number: $w_j^U$ = the upper limit of the criteria weight of a rought number; m = criterion under consideration.

Individual weight values of the criteria are obtained by applying equation (10):

$$RN(Wj) = \left[w_j^L, \; w_j^U\right] = \left[ \frac{\left[q_j^L, \; q_j^U\right]}{\sum\limits_{j=1}^{m} \left[q_j^L, \; q_j^U\right]} \right]$$

(10)

Where $w_j^L$ = the lower limit of the criteria weight of the rough number; $w_j^U$ = the upper limit of the criteria weight of a rough number; $q_j^L$ = the lower limit of the recalculated weight; $q_j^U$ = the upper limit of the recalculated weight; m = criterion under consideration.

## Data collection

To ensure the collection of high-quality data, a team of twelve purposively key experts with extensive knowledge and experience in BI implementation and utilization was assembled. In MCDM research, the validity and reliability of the results depend not on statistical power from a large sample but on the quality, depth of knowledge, and experience of a carefully selected expert panel [92].

A panel of 10–15 experts is a well-established norm in MCDM studies, particularly when employing methods like SWARA and R-SWARA [96]. This approach ensures that the participants possess the requisite expertise to make informed judgments [34]. Given that our panel comprised seasoned professionals from key sectors in Yemen (as detailed in Table 2), this sample size is deemed both appropriate and sufficient to achieve the research objectives.

The participants were selected from a diverse range of large and medium-sized organizations across various sectors within the Yemeni business landscape. Their professional roles varied, including information management officers, data managers, authorization managers, and programmers. The group also represented a range of experience levels, from individuals with less than five years to those with over a decade of expertise.

**Table 2. Experts/ profile information.**

| Feature | Scale | Frequency | Percentage |
|---|---|---|---|
| Gender | Male | 9 | 0.75% |
| | Female | 3 | 0.25% |
| Qualification | BSC | 7 | 0.58% |
| | MSc | 5 | 0.42% |
| Experience in BI | <10 | 5 | 0.42% |
| | 5: 10 | 5 | 0.42% |
| | > 10 | 2 | 0.16% |
| Positions | Information Management Officer | 7 | 0.58% |
| | Programmer | 3 | 0.25% |
| | Data manager | 1 | 0.08% |
| | Authorization manager | 1 | 0.08% |
| Industry | Humanitarian | 2 | 17% |
| | IT services | 4 | 33% |
| | Customs authority | 1 | 8% |
| | Telecom | 2 | 17% |
| | Banking | 3 | 25% |

A detailed overview of the experts' professional profiles is provided in Table 2. To gather expert opinions, a questionnaire was developed and distributed via an online platform, which was used to systematically rank the identified CSFs in descending order of importance. All participants were adult professionals and provided informed written consent prior to participation. Participants were informed that the study was conducted solely for academic purposes, with participation being voluntary and subject to withdrawal at any time without consequence. They were further assured that their responses would remain fully anonymous and confidential.

Furthermore, the data collection was conducted with the formal approval of the Faculty of Computing and Information Technology, Sana'a University (Ref No: 3002). Prior to data collection, all participants were provided with a clear explanation of the study's objectives and their rights as volunteers. The study ensured full anonymity and confidentiality of the participants' identities and their organizational affiliations. Data was used strictly for academic purposes, and participants were informed of their right to withdraw from the study at any time without any repercussions.

## Analysis and results

### Analysis

The relative importance of the CSFs was rigorously determined using the novel Rough-SWARA method. To ensure the reliability of the expert evaluations, we employed the lattice approach proposed by Yazdani, Gonzalez, and Chatterjee (2019) to guide the data collection [94]. This section details the computational analysis used to prioritize the CSFs and identify the most influential factors for successful BI adoption.

This analysis contains seven steps as follows:

**Step 1:** This initial step involved the identification of sixteen CSFs, as presented in Table 1.

**Step 2:** An Expert Panel was initiated. A team of expert individuals was assembled to evaluate the relative importance of the identified CSFs.

**Step 3:** Rough matrix $c_j$ creation: The group rough matrix $c_j$ was developed from the expert evaluations in S1 Appendix (Table A1). Using equation (1), a rough matrix $c_j$ is obtained, as:

$c_1 = [8,1,10,7,10,4,2,9,6,2,8,12]$.

$\underline{\text{Lim}}\,(1) = 1$

$\overline{\text{Lim}}\,(1) = (1/12)\,(8 + 1 + 10 + 7 + 10 + 4 + 2 + 9 + 6 + 2 + 8 + 12) = 6.58$

$\underline{\text{Lim}}\,(2) = (1/3)\,(1 + 2 + 2) = 1.7$

$\overline{\text{Lim}}\,(2) = (1/9)\,(8 + 1 + 10 + 7 + 10 + 4 + 2 + 9 + 6 + 2 + 8 + 12) = 7.09$

$\underline{\text{Lim}}\,(4) = (1/4)\,(1 + 4 + 2 + 2) = 2.25$

$\overline{\text{Lim}}\,(4) = (1/8)\,(8 + 10 + 7 + 10 + 4 + 9 + 6 + 8 + 12) = 8.22$

$\underline{\text{Lim}}\,(6) = (1/5)\,(1 + 4 + 2 + 6 + 2) = 3$

$\overline{\text{Lim}}\,(6) = (1/8)\,(8 + 10 + 7 + 10 + 9 + 6 + 8 + 12) = 8.75$

$\underline{\text{Lim}}$ (7) = (1/6) (1 + 7 + 4 + 2 + 6 + 2) = 3.67

‾‾‾‾

Lim (7) = (1/7) (8 + 10 + 7 + 10 + 9 + 8 + 12) = 9.14

$\underline{\text{Lim}}$ (8) = (1/8) (8 + 1 + 7 + 4 + 2 + 6 + 2 + 8) = 4.75

‾‾‾‾

Lim (8) = (1/6) (8 + 10 + 10 + 9 + 8 + 12) = 9.5

$\underline{\text{Lim}}$ (9) = (1/8) (8 + 1 + 7 + 4 + 2 + 9 + 2 + 8) = 5.22

‾‾‾‾

Lim (9) = (1/4) (10 + 10 + 9 + 12) = 10.25

$\underline{\text{Lim}}$ (10) = (1/11) (8 + 1 + 10 + 7 + 10 + 4 + 2 + 9 + 6 + 2 + 8) = 6.09

‾‾‾‾

Lim (10) = (1/3) (10 + 10 + 12) = 10.67

$\underline{\text{Lim}}$ (12) = (1/12) (8 + 1 + 10 + 7 + 10 + 4 + 2 + 9 + 6 + 2 + 8 + 12) = 6.58

‾‾‾‾

Lim (12) = 12

$c_1^L = (c_1^1 + c_1^2 + c_1^3 + c_1^4 + c_1^5 + c_1^6 + c_1^7 + \ldots + c_1^{12})/ n = 3.63$

$c_1^U = (c_1^1 + c_1^2 + c_1^3 + c_1^4 + c_1^5 + c_1^6 + c_1^7 + \ldots + c_1^{12})/ n = 9.12$

Based on the earlier approximations, the whole matrix RN(Cj) is presented in S1 Appendix (Table A2).

**Step 4:** The previous matrix has to be normalized using equations (2) – (4) in the following way.

The first member, ($s1$), is stated to be equal to one, and the subsequent components of the same matrix are obtained by dividing them by their greatest values.

RN($s1$) = [3.89/ 14.00, 9.12/ 8.59] = [0.28, 1.06]

RN($s2$) = [6.16/ 14.00, 12.37/ 8.59] = [0.44, 1.44]

Similarly, to obtain the matrix, the remaining elements must be calculated. The whole matrix RN(Sj) is presented in S1 Appendix (Table A3).

**Step 5:** Equation (6) is used to create a new matrix; each element in the previous matrix, except the first one, must be added to one. The new matrix RN(Kj) is presented in S1 Appendix (Table A4).

**Step 6:** The elements of the recalculated weight matrix are calculated by applying equation (8) as follows:

$q_1^L = q_{j-1}^L/ k_j^U = q_{13}^L/ k_1^U = 1.00/ 2.06 = 0.48$

$q_1^U = q_{j-1}^L/ k_j^L = q_{13}^U/ k_1^L = 1.00/ 1.28 = 0.78$

$q_2^L = q_{j-1}^L/ k_j^U = q_5^L/ k_1^U = 0.0000.96/ 2.44 = 0.000039$

$q_2^U = q_{j-1}^L/ k_j^L = q_5^U/ k_1^L = 0.04/ 1.48 = 0.03$

It is important to note that j-1 indicates the previous criteria in relation to j. The ranking criteria in step 3 are considered, which means that, for example, the value for the second criterion j-1 makes the fifth criterion, because it is the previous one according to the ranking. The complete matrix RN(Qj) is represented in S1 Appendix (Table A5).

**Step 7:** Finally, in the seventh step, using equation (10), the relative weight values of the criteria are obtained. The example of estimating wj is:

$[w_{13}^L, w_{13}^U]$ = [1.00/4.16, 1.00/1.894945] = [0.24059710, 0.52772]

$[w_1^L, w_1^U]$ = [0.484910/4.16, 0.78/1.894945] = [0.11666799, 0.41287]

Lastly, the relative weights and final ranking of CSFs of BI adoption were determined by employing equation (10). The calculation of matrix $RN(Wj)$ is presented in Table 3. The prioritization of the CSFs according to the Experts' evaluation is presented in Fig 5.

Based on the expert evaluations, the analysis revealed that "Competitive Pressure" was identified as the most critical factor, while "Champion and Balanced Team Composition" was deemed the least critical factor among the evaluated criteria.

## Sensitivity analysis

To ensure the robustness of the results, a sensitivity analysis was performed. Two complementary approaches were applied. First, a leave-one-expert-out procedure was conducted, in which the rankings were recalculated after systematically removing each expert's input. The results demonstrated that the top-ranked factors—competitive pressure, data quality, and clear vision & strategic alignment—remained stable, indicating consistency across expert judgments. Second, a comprehensive sensitivity analysis was accomplished by systematically altering the weight of the top-ranked factor (competitive pressure = 0.38416) from 0.1 to 0.9. A series of incremental addition runs was carried out by scaling the relative weight values of all criteria [97]. The rankings across these runs confirmed that the most influential CSFs maintained their relative positions. This analysis confirmed the robustness and stability of our overall rankings across all tested scenarios. Thus, the results of the study are robust and can be used for decision-making. The relative importance weights of all CSFs using the sensitivity investigation are presented in Table 4. Finally, Table 5 depicts the rankings of CSFs utilizing sensitivity analysis in the study. Moreover, Fig 6 represents the overall variation in the sensitivity analysis.

**Table 3. Weights and ranking of CSFs of BI Adoption.**

| Criterion | Weight | | Crisp = mean value | Rank |
|---|---|---|---|---|
| | **Min** | **Max** | | |
| c13: Competitive Pressure | 0.52771973 | 0.24060 | 0.38416 | 1 |
| c1: Information/data quality | 0.25589669 | 0.18824 | 0.22207 | 2 |
| c10: Clear vision and business strategic Alignment | 0.12064514 | 0.14384 | 0.13224 | 3 |
| c16: Change management | 0.05386514 | 0.11409 | 0.08398 | 4 |
| c11: Organization size | 0.02453579 | 0.08201 | 0.05327 | 5 |
| c14: Regulation | 0.01014762 | 0.06368 | 0.03691 | 6 |
| c3: Compatibility | 0.00421299 | 0.04836 | 0.02628 | 7 |
| c6: IT infrastructure | 0.00172656 | 0.03637 | 0.01905 | 8 |
| c7: Business driven, scalable & flexible technical framework | 0.00074112 | 0.02596 | 0.01335 | 9 |
| c8: Top Management support | 0.00030166 | 0.01883 | 0.00957 | 10 |
| c12: Information sharing culture | 0.00012349 | 0.01337 | 0.00675 | 11 |
| c5: System integration | 0.00005065 | 0.00901 | 0.00453 | 12 |
| c2: Relative advantages | 0.00002075 | 0.00625 | 0.00314 | 13 |
| c9: Adequate resources | 0.00000828 | 0.00440 | 0.00221 | 14 |
| c4: Complexity | 0.00000317 | 0.00308 | 0.00154 | 15 |
| c15: Champion & Balance team Composition | 0.00000121 | 0.00191 | 0.00096 | 16 |

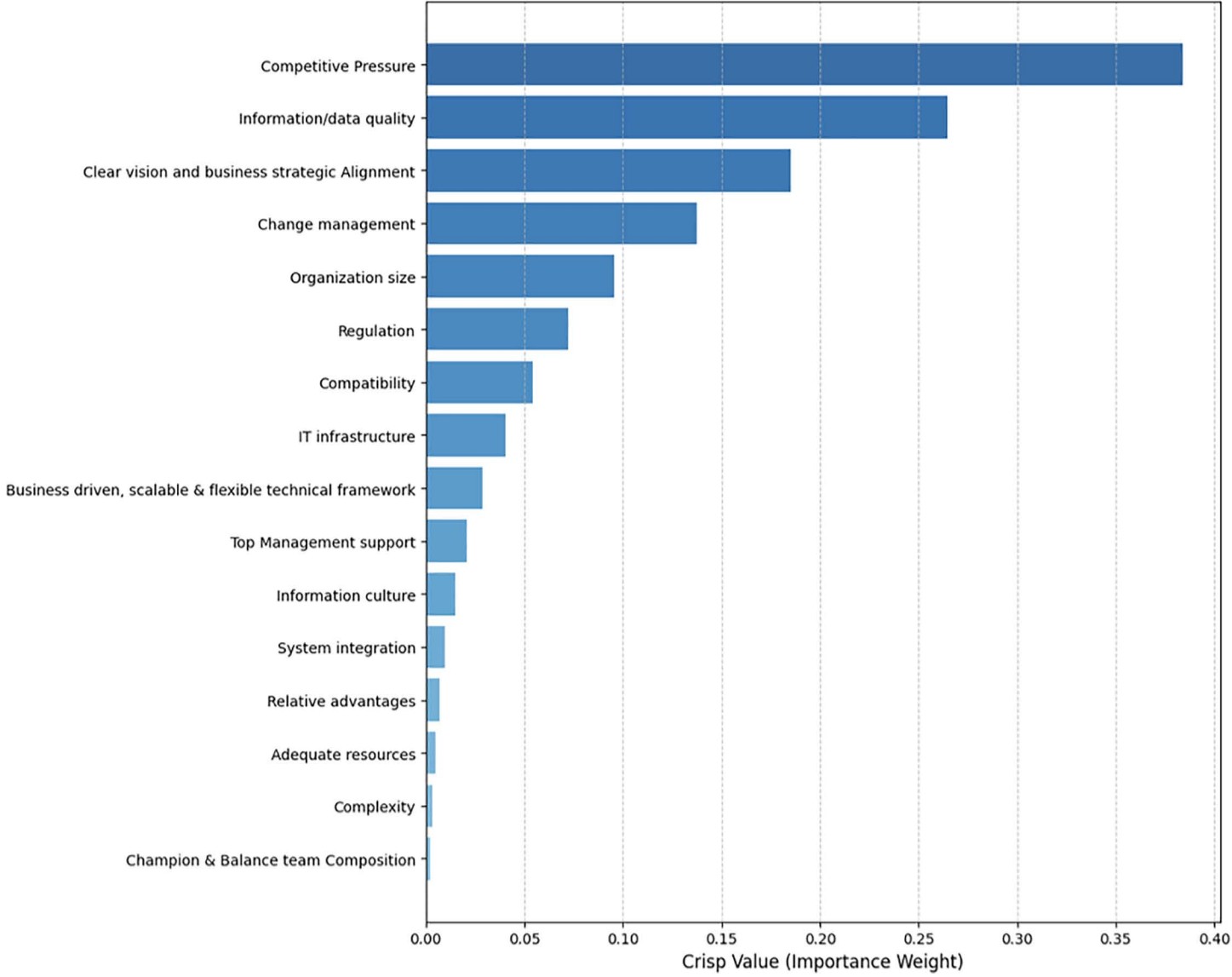

**Fig 5. The CSFs prioritization.**

## Discussion

This study developed a novel TOEP framework by integrating the generally established TOE framework and the BI-specific Yeoh and Koronios [23] model. This framework was evaluated to prioritize the CSFs influencing the adoption of BI systems in the Yemeni context. The R-SWARA method, a multi-criteria decision-making (MCDM) technique, was selected to complete this evaluation for its ability to reduce the number of pairwise comparisons, thereby minimizing both uncertainty and subjectivity in the evaluation process.

The study's findings provide a structured analysis of the priority of sixteen factors. To further systematize the results based on expert opinion, the factors are also grouped into three distinct categories of influence: high, moderate, and low. The prioritized list of CSFs serves as a strategic guide for resource allocation and decision-making.

**Table 4. Sensitivity analysis of CSFs of BI adoption.**

| CSF | Normalized Value | Run 1 (0.1) | Run 2 (0.2) | Run 3 (0.3) | Run 4 (0.4) | Run 5 (0.5) | Run 6 (0.6) | Run 7 (0.7) | Run 8 (0.8) | Run 9 (0.9) | Rank |
|---|---|---|---|---|---|---|---|---|---|---|---|
| c13 | 0.38416 | 0.1 | 0.2 | 0.3 | 0.4 | 0.5 | 0.6 | 0.7 | 0.8 | 0.9 | 1 |
| c1 | 0.22207 | 0.456 | 0.40533 | 0.35467 | 0.304 | 0.25333 | 0.20267 | 0.152 | 0.10133 | 0.05067 | 2 |
| c10 | 0.13224 | 0.27156 | 0.24138 | 0.21121 | 0.18104 | 0.15086 | 0.12069 | 0.09052 | 0.06035 | 0.03017 | 3 |
| c16 | 0.08398 | 0.17245 | 0.15328 | 0.13412 | 0.11496 | 0.0958 | 0.07664 | 0.05748 | 0.03832 | 0.01916 | 4 |
| c11 | 0.05327 | 0.10939 | 0.09724 | 0.08508 | 0.07293 | 0.06077 | 0.04862 | 0.03646 | 0.02431 | 0.01215 | 5 |
| c14 | 0.03691 | 0.0758 | 0.06738 | 0.05895 | 0.05053 | 0.04211 | 0.03369 | 0.02527 | 0.01684 | 0.00842 | 6 |
| c3 | 0.02628 | 0.05397 | 0.04798 | 0.04198 | 0.03598 | 0.02999 | 0.02399 | 0.01799 | 0.01199 | 0.006 | 7 |
| c6 | 0.01905 | 0.03911 | 0.03477 | 0.03042 | 0.02608 | 0.02173 | 0.01738 | 0.01304 | 0.00869 | 0.00435 | 8 |
| c7 | 0.01335 | 0.02741 | 0.02437 | 0.02132 | 0.01827 | 0.01523 | 0.01218 | 0.00914 | 0.00609 | 0.00305 | 9 |
| c8 | 0.00957 | 0.01964 | 0.01746 | 0.01528 | 0.0131 | 0.01091 | 0.00873 | 0.00655 | 0.00437 | 0.00218 | 10 |
| c12 | 0.00675 | 0.01386 | 0.01232 | 0.01078 | 0.00924 | 0.0077 | 0.00616 | 0.00462 | 0.00308 | 0.00154 | 11 |
| c5 | 0.00453 | 0.0093 | 0.00827 | 0.00723 | 0.0062 | 0.00517 | 0.00413 | 0.0031 | 0.00207 | 0.00103 | 12 |
| c2 | 0.00314 | 0.00644 | 0.00573 | 0.00501 | 0.0043 | 0.00358 | 0.00286 | 0.00215 | 0.00143 | 0.00072 | 13 |
| c9 | 0.00221 | 0.00453 | 0.00403 | 0.00352 | 0.00302 | 0.00252 | 0.00201 | 0.00151 | 0.00101 | 0.0005 | 14 |
| c4 | 0.00154 | 0.00317 | 0.00282 | 0.00246 | 0.00211 | 0.00176 | 0.00141 | 0.00106 | 0.0007 | 0.00035 | 15 |
| c15 | 0.00096 | 0.00196 | 0.00174 | 0.00153 | 0.00131 | 0.00109 | 0.00087 | 0.00065 | 0.00044 | 0.00022 | 16 |
| Total | 1 | 1 | 1 | 1 | 1 | 1 | 1 | 1 | 1 | 1 | |

**Table 5. The changes in ranking using sensitivity analysis.**

| CSF | Run 1 (0.1) | Run 2 (0.2) | Run 3 (0.3) | Run 4 (0.4) | Normalized Value (0.384) | Run 5 (0.5) | Run 6 (0.6) | Run 7 (0.7) | Run 8 (0.8) | Run 9 (0.9) |
|---|---|---|---|---|---|---|---|---|---|---|
| c1 | 1 | 1 | 1 | 2 | 2 | 2 | 2 | 2 | 2 | 2 |
| c2 | 13 | 13 | 13 | 13 | 13 | 13 | 13 | 13 | 13 | 13 |
| c3 | 7 | 7 | 7 | 7 | 7 | 7 | 7 | 7 | 7 | 7 |
| c4 | 15 | 15 | 15 | 15 | 15 | 15 | 15 | 15 | 15 | 15 |
| c5 | 12 | 12 | 12 | 12 | 12 | 12 | 12 | 12 | 12 | 12 |
| c6 | 8 | 8 | 8 | 8 | 8 | 8 | 8 | 8 | 8 | 8 |
| c7 | 9 | 9 | 9 | 9 | 9 | 9 | 9 | 9 | 9 | 9 |
| c8 | 10 | 10 | 10 | 10 | 10 | 10 | 10 | 10 | 10 | 10 |
| c9 | 14 | 14 | 14 | 14 | 14 | 14 | 14 | 14 | 14 | 14 |
| c10 | 2 | 2 | 3 | 3 | 3 | 3 | 3 | 3 | 3 | 3 |
| c11 | 4 | 5 | 5 | 5 | 5 | 5 | 5 | 5 | 5 | 5 |
| c12 | 11 | 11 | 11 | 11 | 11 | 11 | 11 | 11 | 11 | 11 |
| c13 | 5 | 3 | 2 | 1 | 1 | 1 | 1 | 1 | 1 | 1 |
| c14 | 6 | 6 | 6 | 6 | 6 | 6 | 6 | 6 | 6 | 6 |
| c15 | 16 | 16 | 16 | 16 | 16 | 16 | 16 | 16 | 16 | 16 |
| c16 | 3 | 4 | 4 | 4 | 4 | 4 | 4 | 4 | 4 | 4 |

The empirical results of this study provide strong validation for the integrated TOEP framework, demonstrating an explanatory capacity that exceeds its foundational models (TOE and Yeoh & Koronios). Notably, the R-SWARA analysis identified 'Competitive Pressure' (Environmental dimension) and 'Change Management' (Process dimension) among the highest-ranking CSFs in the Yemeni context.

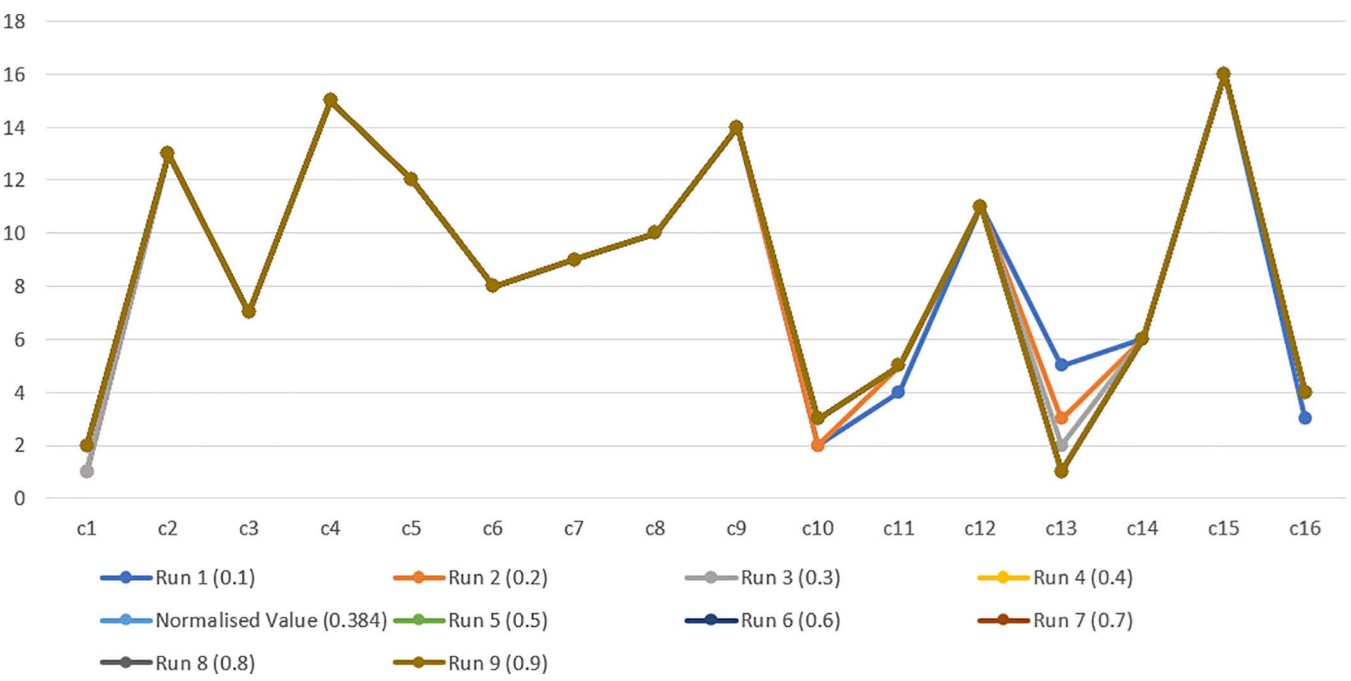

**Fig 6. Overall Variation in the Sensitivity Analysis.**

From a theoretical standpoint, had this study relied solely on the Yeoh and Koronios model, the significant influence of 'Competitive Pressure' would have remained unaccounted for due to that model's 'environmental blindness.' Conversely, if only the TOE framework were employed, the critical role of 'Change Management'—which emerged as a primary driver— would have been obscured within the broad and static 'Organizational' category, failing to capture its dynamic importance as a procedural bridge.

In a volatile environment like Yemen, our findings reveal that the interplay between external market survival (environment) and internal implementation rigor (process) is the true determinant of BI success. Therefore, the TOEP framework does not merely combine factors; it functions as a synergistic analytical tool that captures the high-stakes reality of BI adoption, providing a nuanced understanding that traditional, non-integrated models cannot offer.

## Most significant factors: Organizational and environmental dimensions

The top five factors are ranked according to their weights as follows: competitive pressure (weight = 0.38416)> data quality (weight = 0.22207)> clear vision and business strategic alignment (weight = 0.13224)> change management (weight = 0.08398) and> organization size (weight = 0.05327).

The analysis identifies competitive pressure as the most significant driver of BI adoption. This finding underscores the critical role of the external environment, where the dual motivations of organizational sustainability and the risk of being outpaced by rivals serve as primary catalysts for innovation. Consistent with strategic management literature, BI is recognized as a vital tool for enhancing operational efficiency and leveraging data assets for innovation [98,99]. In today's volatile markets, BI adoption has transitioned from a discretionary advantage to a fundamental necessity for securing a sustained competitive edge [100].

Data quality emerged as the second-most influential factor, aligning with established research that consistently positions data integrity as a top priority [101]. In the big data era, the efficacy of strategic decision-making is entirely contingent

upon the accuracy and reliability of the underlying information [100]. Consequently, data quality remains a central concern in BI projects [102], serving as the bedrock for the trustworthiness and value of the insights generated.

The third most critical factor is clear vision and strategic alignment. Given that BI systems are inherently business-centric, their implementation must be harmonized with the organization's overarching strategic objectives to ensure long-term viability. This finding reinforces the consensus in the literature that organizational competitiveness is contingent upon the seamless alignment of BI initiatives with corporate goals [103–105].

Ranked fourth, change management was identified as the most critical process-related factor. This aligns with prior research suggesting that inadequate change management can lead to organizational disarray and eroded operational efficiency [57,106]. Notably, the high ranking of this factor validates the study's TOEP framework. By expanding the traditional TOE model to include the Process dimension, this study provides a more comprehensive and nuanced view of the BI adoption journey.

Ranking fifth is organization size, which plays a vital role in adoption readiness for any new technology by serving as a key determinant in providing the necessary resources for a successful adoption process [53,91].

**Medium significant factors**

The factors exhibiting moderate influence were ranked according to their relative weights as follows: Regulation (0.03691)> Compatibility (0.02628)> IT Infrastructure (0.01905)> Business-driven, Scalable & Flexible Technical Framework (0.01335)> Top Management Support (0.00957).

The regulation factor ranked sixth in overall importance. This finding aligns with established research [107], highlighting the dual role of regulatory authority: it can either actively catalyze the proliferation of IT or create critical bureaucratic obstacles that stifle technological adoption. Several empirical studies reinforce this, demonstrating that the regulatory landscape is a profound determinant of BI success [108,109].

Compatibility secured the seventh position, suggesting that when BI solutions are seamlessly aligned with an institution's pre-existing processes, the probability of successful adoption increases significantly [14]. Conversely, a lack of technical or operational compatibility tends to truncate system capabilities and diminish the overall value proposition of the BI solution [55].

IT infrastructure was ranked eighth. The flexibility and scalability of an organization's existing technology stack drive adoption through two primary mechanisms: first, by reducing prohibitive initial costs as hardware and networking components are already established [110], and second, by ensuring the system can be expanded to meet evolving organizational requirements [31,111,112]. The business-driven, scalable & flexible technical framework ranked ninth. Its importance lies in two key areas: flexibility is a confirmed capability for overall BI success [23], and the system must be inherently designed to adapt to future expansion requirements to remain useful and viable [82].

Notably, top management support received an unexpectedly low ranking (10th place). This finding markedly contrasts with the dominant literature, which consistently positions leadership conviction as a high-impact CSF [23]. The traditional view is clear: without conviction from leadership, an IT innovation will likely fail to be adopted [113]. In the Yemeni context, this suggests a potential "perception gap," where businesses may not yet fully grasp the strategic necessity of BI. This lack of prioritization is further substantiated by the low ranking of relative advantage. This correlation indicates a fundamental lack of awareness regarding BI's transformative benefits among senior leadership, which in turn undermines their willingness to provide the necessary resources and strategic backing.

**Less significant factors: Unique insights from the Yemeni context**

The study reveals a unique set of findings that contrast with traditional BI adoption literature, as several factors typically considered "critical" exhibited surprisingly low importance within the Yemeni market. These factors are ranked as follows:

Information sharing culture (0.00675)> system integration (0.00453)> relative advantages (0.00314)> adequate resources (0.00221)> complexity (0.00154)> champion & balanced team composition (0.00096).

While recognized globally as a vital predictor of BI success [14], information sharing culture ranked only eleventh. This suggests that the analytical culture in Yemen is in a nascent stage. The prevalence of data silos, departmental resistance to transparency, and a general mistrust of data compounded by a lack of formal governance, diminishes the current impact of this factor. Thus, its low ranking is not an anomaly but a reflection of a foundational prerequisite that has yet to be satisfied.

System integration also received an unexpectedly low ranking, appearing to contradict established literature [114,115]. However, this result is consistent with studies identifying integration as a primary hurdle in Yemen [116]. This contradiction arises because many companies in Yemen have yet to fully adopt comprehensive enterprise information systems like ERP, leaving them with fragmented, siloed operations. Many Yemeni firms have not yet implemented comprehensive enterprise systems like ERP, resulting in fragmented, siloed operations [117]. The absence of middleware, weak administrative frameworks, and a shortage of technical expertise [118,119] hinder the seamless data flow required for BI functionality, thereby lowering its perceived priority in the early adoption phase [60].

Contrary to the theoretical expectation that perceived benefits accelerate adoption [120], relative advantages ranked low. This aligns with findings in similar contexts [85,121] but diverges from others [52,122]. In Yemen, BI systems are often underutilized and in their infancy; consequently, organizations have yet to realize their full strategic value. The lack of robust analytics culture further obscures the transformative impact of BI, leading to a diminished awareness of its inherent advantages.

Finally, complexity, adequate resources, and champion & balanced team composition were ranked as the least significant factors. While these are often cited as major barriers [67,123], their low ranking here suggests that Yemeni organizations are currently adopting simpler, "off-the-shelf" solutions. Such implementations minimize the need for high-level customization, massive resource pools, or specialized project leadership.

Overall, these findings suggest that the drivers and barriers of BI adoption in a volatile, developing economy like Yemen are fundamentally distinct from those in developed nations. In this context, the primary focus is on environmental survival (competitive pressure) and initial implementation hurdles, rather than the advanced organizational and process-related factors that define mature BI landscapes.

## Implications and limitations

In nascent and under-researched contexts like Yemen, the investigation of **Business Intelligence (BI)** adoption is paramount for generating essential localized knowledge and providing a strategic roadmap for future digital initiatives. The primary contribution of this study lies in its unique perspective from a conflict-affected, developing economy, demonstrating that adoption drivers and barriers are not universal phenomena but are profoundly contingent upon the local institutional and economic environment. By focusing on the Yemeni landscape, this research illuminates how the specific challenges faced in such volatile contexts diverge significantly from the established norms of developed nations. Consequently, this work offers a vital and distinctive contribution to the global BI discourse, challenging the generalizability of existing frameworks and advocating for a more contextualized approach to technological innovation.

## Theoretical implications

This research offers a novel conceptual model by integrating the TOE (Technology-Organization-Environment) framework with the Yeoh and Koronios [23] model to validate the CSFs of BI adoption. This enriched theoretical lens is expected to contribute to the growing body of literature on BI by highlighting the contextual diversity of technology adoption models, especially in emerging economies.

The theoretical novelty of this study lies in the development of the TOEP framework, which functions as a contextual theoretical bridge between general adoption theory and BI-specific implementation needs. Unlike existing models that suffer from either 'generic oversimplification' (TOE) or 'environmental blindness' (Yeoh & Koronios), our framework provides a synergistic lens that reconciles external volatility with internal procedural rigor. By elevating 'Process' to a standalone dimension, this research moves beyond mere model synthesis toward contextual theory building. It establishes that in emerging economies like Yemen, the success of complex systems is determined by the dynamic interplay between environmental pressures and structured implementation processes—a nuance previously uncaptured in consolidated adoption literature.

Furthermore, this study represents an initial effort to utilize the R-SWARA method for prioritizing BI CSFs. This methodological contribution can serve as a foundational step for future research. This approach establishes a valuable foundation for future inquiries. We call for researchers to extend the use of the proposed TOEP model and its CSFs, validating their applicability in different sectors and environments to significantly enrich the collective understanding of technology adoption.

The study also provides unique context-specific insights. Unlike much of the existing literature from developed countries, which often ranks factors like information sharing culture, system integration, and top management support very highly, our findings show these to be of surprisingly low importance in the Yemeni context. This suggests that the hierarchy of critical success factors for BI adoption can be significantly influenced by local market conditions, infrastructure maturity, and organizational readiness.

The low ranking of typically significant factors, including information sharing culture, system integration, and relative advantages, highlights a fundamental theoretical gap. Our findings suggest that in developing economies like Yemen, these factors are currently not key drivers of adoption but rather act as significant barriers. This reframing provides a new lens for viewing technology adoption in similar contexts. It also suggests that the lack of prerequisites will diminish the comparative relative advantages of these systems, impacting traditionally high-ranking factors, such as top management support. Ultimately, this research offers a nuanced understanding of BI adoption, strongly advocating for the use of more comprehensive frameworks and context-specific analyses to accurately reflect the realities of diverse global markets.

## Practice implications

Prioritizing BI systems' CSFs provides a clear roadmap for successful system implementation and usage. This roadmap and guidance help business leaders, IT managers, project managers, and policymakers to strategically direct significant investments toward the most crucial areas, ensuring the successful adoption of BI. Therefore, these guidelines are divided into three categories: 1) guidance for decision-makers and business managers; 2) guidance for BI providers and IT managers; 3) guidance for policymakers.

## Guidance for decision-makers and business managers

The high prioritization of competitive pressure, strategic vision, and change management underscores that Business Intelligence (BI) adoption in Yemen is primarily catalyzed by external market exigencies and internal strategic imperatives. To navigate this landscape successfully, business leaders must shift their perspective, treating BI not merely as a peripheral technical tool but as a core strategic asset essential for market survival and competitive positioning. This shift requires that BI initiatives be governed by a clear vision directly integrated with the organization's overarching business objectives and Key Performance Indicators (KPIs). By anchoring BI efforts to organizational strategic objectives, leadership can ensure that data-driven insights translate into measurable value, thereby maximizing the Return on Investment (ROI) and securing long-term top management support.

Furthermore, the significant weight assigned to change management necessitates a comprehensive, human-centric implementation strategy. In the Yemeni context, mismanagement of this transition can lead to operational disarray;

therefore, managers must prioritize robust employee training and proactive awareness campaigns to articulate the tangible benefits of BI. Actively managing resistance to change through a culture of transparency and inclusiveness is critical for facilitating smoother transitions and ensuring high user adoption rates. The effectiveness of the BI system is ultimately dependent on the willingness of the workforce to integrate these tools into their daily workflows, making the "process" dimension as vital as the technology itself.

Moreover, our findings show that typically crucial factors like information sharing culture and system integration are not yet the primary drivers for BI adoption in Yemen; rather, they are significant barriers. Therefore, businesses must first focus on the analysis of these gaps. To cultivate a supportive information sharing culture, managers and BI adopters should focus on two key actions: first, they must raise awareness about the value of data-driven decision-making to establish trust in organizational data and actively combat mistrust; and second, they need to actively foster an analytical culture that encourages robust cross-departmental collaboration and data sharing, which is essential for breaking down existing data silos. (Specific technical recommendations for systems integration will be addressed in the section for BI providers and technology managers.)

## Guidance for BI providers and IT managers

The high ranking of data quality highlights its crucial role in the success of BI adoption. To address this, BI providers and IT managers must build a robust data management strategy that combines both organizational policy and modern technology. This begins with establishing a strong data governance framework to ensure data is accurate, consistent, and secure. This is then complemented by leveraging solutions like a data warehouse, data lake, or a Master Data Management (MDM) system to centralize data, eliminate silos, and create a single, reliable source of truth.

The high ranking of compatibility indicates that BI providers and IT managers need to ensure that the solution is not only powerful but also compatible with a company's existing business processes. BI providers and IT managers should choose between two main strategies: either adapt their current business processes to fit the new system or customize the system for seamless alignment with existing procedures. The ideal approach is always the one that achieves best practices with the minimal investment in cost, effort, and time. Moreover, IT managers must ensure that the IT infrastructure can support the adoption of BI technologies. This includes large capacies and superior analytical capabilities to meet the needs of BI systems. They also must ensure that the system's technical framework is both flexible and scalable to meet the ever-changing business requirements and ensure system quality and usability.

The relatively low weight of top management support highlights the importance of raising awareness of BI potential. Therefore, BI providers and IT leaders, such as Chief Technology Officers (CTOs) and Chief Information Officers (CIOs), must proactively raise awareness of BI's relative advantages. This effort is to ensure that managers and decision-makers are aware of the strategic capabilities of BI technologies to ensure their continuous and full support for these innovative systems. The low ranking of the relative advantages factor strongly supports the previous proposal, as it confirms that the advantages of BI systems are neither fully understood nor realized yet.

Finally, the low weighting of systems integration confirms that this area faces significant, persistent challenges. Overcoming these challenges requires a strategic focus across four dimensions: strategy, processes, technology, and people. Organizations must first develop a comprehensive integration strategy that aligns with overarching business goals and involves all key stakeholders, directly addressing any internal administrative resistance to integrating departments and units. Simultaneously, they must reengineer and optimize business processes to fully leverage the best practices and integrated workflows. On the technology front, companies should use flexible solutions like application programming interfaces (APIs) and middleware to create a scalable, integrated framework. Implementing robust integration methods, such as extract, load, transform (ELT) pipelines, is crucial to ensuring a single, authoritative data repository. Finally, it is essential to invest in training and expertise to effectively bridge the existing skills gap required for managing complex integrated systems.

## Guidance for policymakers

Given the immaturity of the BI market and the unique challenges identified in Yemen, policymakers and government bodies have a crucial, strategic role in fostering adoption by addressing systemic barriers. This requires a three-pronged approach: first, investing in robust national digital infrastructure (like high-speed networks and secure cloud resources) to provide the necessary technological backbone; second, actively promoting data literacy and analytical skills throughout the workforce and educational systems to correct the poor information sharing culture identified as a barrier; and third, establishing clear, supportive regulatory and legal frameworks that encourage data governance and technological investment. By undertaking these structural efforts, policymakers can create a conducive and fertile environment for future BI adoption nationwide.

## Limitations and future work

This study's limitations, particularly concerning the nascent state of BI adoption in unstable developing economies like Yemen, are better viewed as inspiring opportunities for future research. While BI systems are gaining rapid traction in these regions, their full potential remains untapped, creating a rich area for scholarly contribution. However, this immaturity poses practical research challenges, notably in securing a large pool of local experts. Although the selection of twelve experts strictly aligns with MCDM protocols and R-SWARA requirements, expanding this sample in future studies through large-scale empirical surveys could further strengthen the external validity of the results. To address this, future studies could employ large-scale empirical surveys or longitudinal case studies to further validate the TOEP framework across diverse organizational settings. This would be particularly valuable in testing the framework's adaptability in other emerging or conflict-affected economies. Furthermore, regarding the methodology, the R-SWARA approach was strategically employed for its native ability to manage expert subjectivity and vagueness through lower and upper approximations. While the study prioritized these 'rough number' approximations over traditional consensus measures like Kendall's W, we recognize that incorporating formal inter-rater agreement protocols in future research could provide an additional layer of methodological refinement. Furthermore, since developed countries typically possess infrastructures more receptive to IT innovation, developing nations often face unique forms of socio-technical resistance. Consequently, there is a clear need for mixed-methods research to investigate BI adoption within these complex, constrained contexts. Future studies should aim to provide deeper qualitative insights into the 'Process' dimension' critical component of our framework to capture how these success factors evolve as BI systems transition from initial adoption to full organizational integration.

## Conclusion

This study provides a foundational, quantitative investigation into Business Intelligence (BI) adoption within the challenging organizational landscape of Yemen. The research offers two primary contributions: the development of the novel TOEP framework and the application of the rigorous Rough-SWARA (R-SWARA) method to prioritize Critical Success Factors (CSFs). By identifying competitive pressure, data quality, clear vision, and change management as the most significant drivers, this analysis underscores that BI adoption in this environment is catalyzed by external market urgency and robust internal strategic management reflecting the high-stakes nature of a developing economy.

Conversely, the relatively low ranking of relative advantages indicates that the strategic benefits of BI have yet to be fully realized by top management, subsequently hindering their support. Crucially, these findings highlight a significant theoretical shift for technological adoption in developing economies. Factors traditionally viewed as primary drivers globally—such as information-sharing culture and system integration—currently function as profound barriers in Yemen. This divergence suggests that Yemeni organizations face systemic, economic, and technological hurdles that are largely a consequence of the ongoing conflict and political instability.

Ultimately, the validated TOEP model and its prioritized CSFs offer a context-specific roadmap for both researchers and practitioners. By moving beyond traditional, non-integrated models, this study provides the necessary insights to guide future research and assist Yemeni organizations in leveraging BI as a critical tool for strategic survival and competitive advantage.

## Supporting information

**S1 Appendix. R-SWARA Calculation Procedures.** This file contains the detailed steps of the Rough SWARA method, including expert initial preferences, calculated rough boundaries, and final weight derivations.
(DOCX)

**S2 Appendix. Research Questionnaire.** The survey instrument was used to collect expert judgment.
(DOCX)

## Author contributions

**Conceptualization:** Amira Taha Al-Adimi.

**Formal analysis:** Amira Taha Al-Adimi.

**Investigation:** Amira Taha Al-Adimi.

**Methodology:** Amira Taha Al-Adimi.

**Validation:** Amira Taha Al-Adimi.

**Writing – original draft:** Amira Taha Al-Adimi.

**Writing – review & editing:** Mokhtar Mohammed Ghilan, Walid Shaher Yousef, Abdullatif Ghallab.

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
