## [Decision Letter · Decision Letter 0]

5 Sep 2025

Dear Dr.  Al-Adimi,

Thank you for submitting your manuscript to PLOS ONE. After careful consideration, we feel that it has merit but does not fully meet PLOS ONE’s publication criteria as it currently stands. Therefore, we invite you to submit a revised version of the manuscript that addresses the points raised during the review process.

We look forward to receiving your revised manuscript.

Kind regards,

Muhammad Faheem, PhD

Academic Editor

PLOS ONE

Journal Requirements:

3. We note that your Data Availability Statement is currently as follows: All relevant data are within the manuscript and in Supporting Information files.

**Comments to the Author**

1. Is the manuscript technically sound, and do the data support the conclusions?

Reviewer #1: Partly

Reviewer #2: No

2. Has the statistical analysis been performed appropriately and rigorously?

Reviewer #1: No

Reviewer #2: No

3. Have the authors made all data underlying the findings in their manuscript fully available?

Reviewer #1: Yes

Reviewer #2: Yes

4. Is the manuscript presented in an intelligible fashion and written in standard English?

Reviewer #1: No

Reviewer #2: No

Reviewer #1: The paper presents a potentially valuable contribution but requires substantial revision to meet the journal’s quality standards. The authors are encouraged to revise the manuscript with attention to language clarity, methodological rigor, and contextual interpretation.

Reviewer #2: The book has obvious divisions for the introduction, methods, findings, and discussion, however the wording is repetitious and wordy, making it hard to read. The abstract and introduction give context but fail to identify the research need or the study's distinctive contribution. Thus, whether the dataset, analytical approach, or contextual application is innovative is unclear. While broad, the literature review lists past research without critically synthesizing them or connecting the new study to gaps. A conceptual framework or more analytical approach to past research might bolster the study's reasoning.

The technique is detailed and backed by formulae, but it may be too sophisticated for a general audience. The weighting methods and data source selection are not justified, which reduces trust in the approach's robustness. The lack of robustness tests and sensitivity analysis raises doubts about the findings' credibility. The data are thorough, but the big tables make them hard to analyze, and many of the conclusions support what was predicted rather than give new insights. More persuasive results would need better visualization and a focus on novel or surprising discoveries.

The discussion part fails to relate the results to theories or frameworks, limiting its academic value. Policy implications are vague and unspecific, generally saying “improve coordination” or “increase efficiency.” To bring value, suggestions should be targeted and results-based. Presentation and linguistic flaws weaken the paper. Many words are unclear, technical jargon is utilized without explanation, and tables and figures lack captions and narrative coherence.

The article has technical skill and systematic analysis but lacks intellectual depth, creativity, and practical significance. The authors could clarify the research gap and contribution, explain methodological choices, emphasize relevant discoveries rather than anticipated results, and improve theoretical interpretation and policy suggestions. By improving clarity, innovation, and practicality, the paper might be much enhanced.

**Do you want your identity to be public for this peer review?** For information about this choice, including consent withdrawal, please see our Privacy Policy

Reviewer #1: No

Reviewer #2: **Yes:** Associate prof Mahadi Hasan Miraz

---

## [Author Response · Author response to Decision Letter 1]

16 Oct 2025

Editore Comments:

1. Journal Requirement 1 – Style & Formatting: Ensure manuscript follows PLOS ONE style.

- Reformatted manuscript, checked against PLOS ONE templates.

2. Journal Requirement 2 – Ethics: Clarify participant consent type and process.

- Revised Ethics Statement stated: "All participants were adult professionals and provided informed written consent prior to participation. Participants were informed that the study was conducted solely for academic purposes, with participation being voluntary and subject to withdrawal at any time without consequence. They were further assured that their responses would remain fully anonymous and confidential."

3. Journal Requirement 3 – Data Availability: Confirm minimal dataset.

- We added new section: Data Availability Statement: All relevant data are within the manuscript. The minimal dataset, including the expert rankings, R-SWARA inputs, calculations, outputs, and the values underlying the tables and figures, has been provided inside the manuscript and the appendix. This data is sufficient to replicate all analyses and findings reported in this paper.

4. Journal Requirement 4 – Data Availability Statement of the Submission.

- To address the concern regarding the data sharing plan: Data Location: We confirm that no separate data files or public repository submissions are required. All necessary underlying data for this study—specifically the expert ratings, all R-SWARA calculation data, and sensitivity analysis—are provided in their entirety and in detail within the main manuscript file and in Appendix 1.

- Compliance: As the data is an integral part of the article's appendix, it will be freely and openly accessible to all readers immediately upon publication, fully satisfying the open data requirement without the need for a separate deposition process.

- Submission Update: We have now updated the Data Availability Statement within the submission form to reflect this precise location and confirm immediate open access upon acceptance.

Rewires comments:

1. Reviewer #1 #2: The manuscript must describe a technically sound piece of scientific research with data that supports the conclusions. Experiments must be conducted rigorously, with appropriate controls, replication, and sample sizes. The conclusions must be drawn appropriately based on the data presented.

Strengthen the demonstration of scientific rigorous to explicitly justify its soundness:

1. Methodological Rigor & Controls: The R-SWARA method was selected for its specific subjectivity and uncertainty in handling expert judgment.

2. Sample Size (Expert Panel): The sample size of 12 experts is appropriate and well-justified for an MCDM study. In MCDM research, the focus is on the depth of knowledge of a carefully selected expert panel rather than a large, statistical sample. Our panel (detailed in Table 2) was purposively selected to include professionals with extensive, hands-on experience in BI from a diverse range of key sectors in Yemen, ensuring the data is of high quality and contextually relevant.

3. Replication: To address replication, we have now included a comprehensive Sensitivity Analysis (Section 5.2). This analysis systematically tests the stability of our results by varying the weight of the top-ranked factor, demonstrating that the overall ranking of CSFs remains robust. This acts as a form of computational replication, confirming the reliability of our findings.

4. Data Supporting Conclusions: The Discussion (Section 6) has been substantially revised to meticulously link each key finding (e.g., the high rank of Competitive Pressure, the low rank of Top Management Support) directly back to the results data (Table 3) and to existing literature. We explicitly explain these findings in the unique context of Yemen's developing economy, ensuring all conclusions are firmly grounded in and justified by the data presented.

2. Reviewer #1: Paper has potential but needs improvement in language, methodological rigor, and contextual interpretation.

We have undertaken a comprehensive revision of the manuscript's language and structure. This included:

- Editing to enhance clarity and reduce wordiness.

- Restructuring of the methodology section to provide clearer justification for R-SWARA selection and robustness tests.

- Strengthened contextual interpretation in the discussion by linking findings to Yemen's specific socio-economic context.

3. Reviewer #2 – 1: Abstract & Introduction: Too general, lacks research gap and contribution.

Rewritten to emphasize:

- Addresses the empirical research gap in developing economies by investigating Critical Success Factors (CSFs) for BI adoption specifically in Yemen.

- Developing a novel integrated TOEP framework (TOE + Yeoh & Koronios model) and applies the rigorous R-SWARA method.

- Providing actionable, contextualized findings, identifying competitive pressure, data quality, clear vision, and change management as key drivers, while highlighting information sharing culture and system integration as unique challenges in the Yemeni context, which contradicts patterns in stable economies.

4. Reviewer #2 – 2: Literature Review: Descriptive, not analytical; lacks synthesis.

Reorganized into analytical synthesis:

1. Shift from Description to Analysis

The review avoids merely stating what previous authors found. Instead, it analyzes the limitations and strengths of the foundational models:

- Analysis of TOE: Acknowledging the framework's "comprehensive, holistic, and flexible perspective" (strength) but immediately pivots to its "generality" as a major limitation for a "complex, context-specific system like BI," noting its lack of specificity for process factors (weakness).

- Analysis of Yeoh and Koronios Model: Praising its focus on BI's "unique complexities" (strength) but highlights its "major theoretical shortfall": the neglect of the broader environmental context (weakness).

2. Achieving Synthesis and Justification

Synthesis means blending elements to form a new, coherent argument.

- Synthesis of Gaps: The text explicitly synthesizes the weaknesses of the two models, arguing that the TOE is too generic (missing process detail) while the Yeoh and Koronios model is too internally focused (missing environmental detail).

- Logical Framework Justification: This synthesis leads directly and logically to the study's primary theoretical contribution: the new Technology-Organization-Environment-Process (TOEP) framework. The TOEP model is presented not as a random addition, but as the necessary solution to the combined theoretical shortcomings identified.

3. Clear Research Gap Statement

The argument concludes by firmly grounding the theoretical gap in the research context:

- The need for the TOEP framework is tied to the complexity of the Yemeni context, where both "external pressures (environment) and internal implementation (process) are critical.

- The review ends by stating that the integrated framework and methodology will "effectively investigate the CSFs for BI adoption in the under-researched and challenging context of Yemen."

5. Reviewer #2 – 3: Methodology: Lacks justification for weighting and robustness tests.

-Expanded methodology section with detailed justification for R-SWARA selection over AHP/FAHP, highlighting its advantages in handling expert judgment uncertainty through rough numbers

- Added sensitivity analysis (Section 5.2, Table 9) testing robustness by varying the top factor's weight from 0.1 to 0.9, confirming ranking stability

- Enhanced description of sample size and sexpert selection criteria and profile diversity (Table 2)

6. Reviewer #2 – 4: Results: Hard to follow; large tables dominate; little novelty.

- Improved Readability: Condensed large tables: Shifted detailed tables (e.g., raw expert rankings, R-SWARA calculations) to Appendix A, keeping only the essential summary tables in the main body. The "Results" section in the main paper (Section 5) becomes much less cluttered and easier to follow. The core findings are now presented more clearly.

- Enhanced Presentation/Added Visualization: Incorporated graphs and figures (e.g., Figure 4: CSF Prioritization, Figure 5: Sensitivity Analysis) to improve visual comprehension and provide an immediate, visual understanding of the CSF rankings and the stability of the results, directly combating the "hard to follow" critique.

- Emphasized Novelty: Emphasized the Yemeni-context-specific insights in the Discussion and Conclusion sections to underscore the study's unique contribution. The discussion (Section 6) explicitly contrasts their findings from Yemen with established literature. Key contextual insights include competitive pressure as the top driver, reflecting a survivalist market. Top Management Support and Relative Advantages ranked surprisingly low, suggesting a lack of awareness or realization of BI's value. Information culture and system integration are identified not as drivers but as fundamental barriers in the Yemeni context, a significant theoretical insight.

7. Reviewer #2 – 5: Discussion: Weak theoretical linkage and vague policy implications.

- Enhanced Theoretical Foundation: By deliberately weaving the findings back to established theories, the authors have elevated the discussion from a mere presentation of results to a meaningful scholarly interpretation.

- Actionable Implications: The paper now clearly separates guidance for:

1. Decision-Makers & Business Managers

2. BI Providers & IT Managers.

3.Policymakers.

This structure ensures that the implications are contextualized and practical, offering clear next steps for different stakeholders operating in an environment such as Yemen.

8. Reviewer #2 – 6: Many words are unclear, technical jargon is utilized without explanation, and tables and figures lack captions and narrative coherence.

-Language and Jargon: We have reviewed the entire text to simplify unclear language and define technical terms upon their first use (e.g., "R-SWARA," "TOEP framework"). The manuscript has also been proofread to enhance its overall fluency and academic tone.

-Tables and Figures: All tables and figures now have clear, descriptive captions that explain their purpose and content.

---

## [Decision Letter · Decision Letter 1]

26 Dec 2025

Dear Dr. Al-Adimi,

Thank you for submitting your manuscript to PLOS ONE. After careful consideration, we feel that it has merit but does not fully meet PLOS ONE’s publication criteria as it currently stands. Therefore, we invite you to submit a revised version of the manuscript that addresses the points raised during the review process.

We look forward to receiving your revised manuscript.

Kind regards,

Kao-Yi Shen, Ph.D.

Academic Editor

PLOS One

Journal Requirements:

Reviewers' comments:

Reviewer's Responses to Questions

**Comments to the Author**

Reviewer #2: All comments have been addressed

Reviewer #3: (No Response)

Reviewer #4: All comments have been addressed

2. Is the manuscript technically sound, and do the data support the conclusions?

Reviewer #2: Partly

Reviewer #3: (No Response)

Reviewer #4: Partly

3. Has the statistical analysis been performed appropriately and rigorously?

Reviewer #2: N/A

Reviewer #3: (No Response)

Reviewer #4: N/A

4. Have the authors made all data underlying the findings in their manuscript fully available?

Reviewer #2: Yes

Reviewer #3: (No Response)

Reviewer #4: No

5. Is the manuscript presented in an intelligible fashion and written in standard English?

Reviewer #2: Yes

Reviewer #3: (No Response)

Reviewer #4: No

Reviewer #2: Several elements of the research might benefit from more review. The theoretical contribution, albeit lucid, is somewhat constrained to a synthesis of previous models rather than representing a wholly novel theoretical advancement. Research would be more robust if it demonstrated the empirical superiority of the TOEP framework or its enhanced explanatory capacity relative to its foundational models. Furthermore, although the rationale for using a cohort of twelve experts adheres to MCDM standards, the limited sample size constrains the external validity of the results. Gathering triangle data sources, such case studies or surveys, might enhance the generalizability of conclusions for further study. The methods section, although thorough, may be rather technical for those unacquainted with decision-making models; reducing the mathematical presentation or include a graphic summary would enhance accessibility. Additionally.

The manuscript has substantial improvements in design, clarity, and analytical depth relative to earlier versions. It significantly contributes to the subject by examining a relatively underexplored setting and using rigorous analytical methods to identify and prioritize the drivers of business intelligence adoption. This research, with its refined theoretical definition, streamlined presentation, and modest linguistic enhancements, has significant potential for publication in PLOS ONE as a valuable technical and contextually relevant addition to the understanding of business intelligence uptake in emerging countries.

Reviewer #3: The manuscript is scientifically sound and contributes meaningfully to BI adoption research. Only minor stylistic and interpretive refinements are needed before final acceptance

‏-While the manuscript has improved, several sections especially in the literature review and methodology would benefit from an additional language polish to ensure smoother flow and reduce repetition.

- The link between each CSF ranking and its implications in the Yemeni context could be strengthened further. Some interpretations remain descriptive rather than analytical.

- Ensure consistent referencing style throughout the manuscript.

- ‏Check table numbering; ensure alignment with the narrative.

-Some technical terms should be defined once and not repeated excessively.

Reviewer #4: After reviewing this manuscript, I still think you have to rewise thus manuscript following sections;

Comment-1:

In general, please note that proofreading the paper may be beneficial.

Comment- 2:

Introduction is poorly written. The flow of writing is missing throughout this section. Some critical shortfalls of this section are: (i) Research gap is not clear. Try to write the research gap more clearly and specifically. (i) Add relevent and most current years refrences. (iii) The novelty of the current study is not included in this section. Explain how this work is different from other works done in this field; that is the uniqueness of this study.

Comments 3:

Overlap between TOE and Yeoh & Koronios: The integration of TOE and Yeoh & Koronios is not fully justified. Both models already share organizational and technological dimensions. The added “Process” dimension could be argued as a subset of “Organization” in TOE. A clearer philosophical and theoretical justification for the synthesis is needed.

Comments 4:

The authors describe consent but do not name an Institutional Review Board (IRB) or provide an approval number. This is a mandatory requirement for PLOS ONE and must be added.

The response mentions Appendix 1 with detailed tables, but it is not included in the provided PDF excerpt. The journal must ensure the appendix is submitted and accessible.

While expert judgment is central to R-SWARA, no measure of agreement among experts (e.g., Kendall’s W) is reported. This is a methodological weakness that should be acknowledged or addressed.

The panel includes humanitarian sector experts. The authors should briefly justify why humanitarian experts are relevant to a business intelligence adoption study in Yemen.

In some places, “information culture” is used, elsewhere “information-sharing culture.” Standardize terms throughout.

Some figures and tables are referenced in the text but are not included in the submitted excerpt (e.g., Fig 1, Fig 2). Ensure all are present and correctly numbered.

**Do you want your identity to be public for this peer review?** For information about this choice, including consent withdrawal, please see our Privacy Policy

Reviewer #2: **Yes:** Associate professor Dr Mahadi Hasan Miraz

Reviewer #3: No

Reviewer #4: No

---

## [Author Response · Author response to Decision Letter 2]

7 Jan 2026

Response to Reviewers

Manuscript ID: PONE-D-25-33603R1

Title: Critical Success Factors Influencing Business Intelligence Adoption: Evidence from Yemen

We sincerely thank the Academic Editor and the reviewers for their thoughtful comments and constructive feedback. Below we provide a detailed, point-by-point response in table format. Revisions are tracked in the file 'Revised Manuscript with Track Changes' and a clean version has also been uploaded.

Reviewer Comment Author Response Location of Revision

Reviewer #2: The theoretical contribution, albeit lucid, is somewhat constrained to a synthesis of previous models rather than representing a wholly novel theoretical advancement.

We thank the reviewer for this observation. We have revised the (i) 'Theoretical Framework' section to explicitly clarify the novelty of our theoretical advancement. While our model (TOEP) is a synthesis, we argue that it represents a systematic resolution of a reciprocal theoretical gap that neither parent model could address independently.

Specifically, the TOE framework, while holistic, lacks the 'procedural granularity' required for BI’s unique implementation lifecycle (as noted in Section 2.2). Conversely, the Yeoh & Koronios model, while process-centric, suffers from 'environmental blindness' by neglecting external pressures like competitive and regulatory factors (as noted in Section 2.3).

The novelty of the TOEP framework lies in its ability to reconcile these two perspectives into a single, unified lens specifically adapted for volatile, emerging contexts like Yemen. In such environments, the interplay between external instability (Environment) and internal change management (Process) is critical. This 'Contextual Theory Building' provides a more nuanced explanatory power than either model alone, thus offering a refined theoretical tool for future BI research.

And revised the (ii) Theoretical Implications section to emphasize that our contribution represents 'Contextual Theory Building'. We argue that the novelty lies in redefining the relationship between external volatility (Environment) and internal dynamics (Process). We have highlighted that in emerging and conflict-affected contexts like Yemen, this 'Process-centric' synthesis is a necessary theoretical advancement to move beyond generic IT adoption models, offering a refined analytical tool previously uncaptured in consolidated literature. (i)Theoretical Framework: Paragraph 5, Lines 214-218.

(ii) Theoretical Implications: Paragraph 2 Lines 745-748.

Reviewer #2: Research would be more robust if it demonstrated the empirical superiority of the TOEP framework or its enhanced explanatory capacity relative to its foundational models. To demonstrate the enhanced explanatory capacity of the TOEP framework, we have updated the Discussion section to show how the empirical results (specifically the high ranking of 'Environmental' and 'Process' factors) justifies the integrated approach. Without this synthesis, the critical influence of "Competitive Pressure" (Environment) and "Change Management" (Process) in the Yemeni context would have remained theoretically unexplained by either parent model alone. Discussion: Paragraph 3, 4, 5 Lines 635-647.

Reviewer #2: Furthermore, although the rationale for using a cohort of twelve experts adheres to MCDM standards, the limited sample size constrains the external validity of the results. Gathering triangle data sources, such case studies or surveys, might enhance the generalizability of conclusions for further study. We appreciate the reviewer’s observation regarding the sample size. We have addressed this by: (i) Adding a detailed justification in the data collection section confirming that a panel of 12 experts is not only consistent with R-SWARA and MCDM literature but is often considered optimal for maintaining the quality and depth of expert judgment.

(ii) Including this as a 'Limitation', where we now suggest that future research could employ large-scale surveys or case studies to further validate the TOEP framework across other emerging or conflict-affected economies. (i) Data collection:

Paragraph 2

Lines 516-519.

(ii)Limitation and Future Work:

Paragraph 2

Lines 849-853.

Reviewer #2: The methods section, although thorough, may be rather technical for those unacquainted with decision-making models; reducing the mathematical presentation or include a graphic summary would enhance accessibility. Following the reviewer’s excellent suggestion, we have added Fig 4, which provides a graphic summary of the R-SWARA steps. This flowchart guides the reader through the logical progression from expert ranking to final weight derivation, making the process more accessible to non-technical readers. Furthermore, the detailed mathematical proofs and intermediate calculation tables (Steps 3-7) have been moved to S1 Appendix to improve the flow of the main manuscript. Research methodology, R-SWARA method:

the Figure is added at the end of the section.

Reviewer #3: While the manuscript has improved, several sections especially in the literature review and methodology would benefit from an additional language polish to ensure smoother flow and reduce repetition. We thank the reviewer for this suggestion. The entire manuscript has undergone a comprehensive language polishing and proofreading process. We have reviewed the manuscript to improve the narrative flow, eliminate redundant technical terms. The whole manuscript

Reviewer #3: The link between each CSF ranking and its implications in the Yemeni context could be strengthened further. Some interpretations remain descriptive rather than analytical. We appreciate this valuable feedback. We have significantly revised the implication section deepen the analysis of our findings within the Yemeni context. For instance, we now explicitly discuss how the prioritized factors (such as data quality and management support) are directly influenced by the current economic and operational challenges in Yemen. This shift from descriptive to analytical interpretation provides more practical insights for stakeholders in the region.

Reviewer #3: Ensure consistent referencing style throughout the manuscript. We have meticulously reviewed the reference list to ensure full compliance with PLOS ONE’s formatting guidelines. The whole manuscript

Reviewer #3: Check table numbering; ensure alignment with the narrative. All figures/tables have been re-verified, re-numbered, and uploaded as high-resolution files. The whole manuscript

Reviewer #3: Some technical terms should be defined once and not repeated excessively. We appreciate this suggestion to improve the manuscript's readability. We have conducted a thorough review of the text to ensure that technical terms (such as 'Rough SWARA', 'Information-sharing culture', and 'TOEP framework') are defined clearly upon their first mention and used consistently thereafter. We have removed redundant definitions and excessive repetitions. The whole manuscript

Reviewer #4: In general, please note that proofreading the paper may be beneficial. We sincerely appreciate this suggestion. The entire manuscript has undergone a comprehensive proofreading and language editing process. We have corrected grammatical errors, improved sentence structure, and ensured that technical terminology is used consistently and accurately throughout the text. Special attention was given to the 'Methods' and 'Results' sections to ensure clarity and professional academic flow. The whole manuscript

Reviewer #4: Introduction is poorly written. The flow of writing is missing throughout this section. Some critical shortfalls of this section are: (i) Research gap is not clear. Try to write the research gap more clearly and specifically. (i) Add relevant and most current years references. (iii) The novelty of the current study is not included in this section. Explain how this work is different from other works done in this field; that is the uniqueness of this study. The Introduction has been entirely rewritten to address these concerns. We have: (i) specifically defined the research gap regarding BI in conflict-affected environments.

(ii) clearly articulated the uniqueness of the TOEP framework. And the R_SWARA employment for CSF prioritization. (i) introduction:

Paragraph 2, 3, 4.

Lines 35-52.

(ii) introduction:

Paragraph 5,6,7.

Lines 53-81.

Reviewer #4: Overlap between TOE and Yeoh & Koronios: The integration of TOE and Yeoh & Koronios is not fully justified. Both models already share organizational and technological dimensions. The added “Process” dimension could be argued as a subset of “Organization” in TOE. A clearer philosophical and theoretical justification for the synthesis is needed. We appreciate the reviewer’s request for a deeper theoretical justification. We have addressed this by refining the manuscript in two key locations to clarify why 'Process' is treated as an independent dimension:

1. First, in the Theoretical Development section: We have integrated a detailed rationale based on the work of Yeoh and Koronios (2010). We clarified that unlike static organizational traits, 'Process' represents the dynamic, evolutionary journey of BI implementation. We explicitly added that BI requires a dedicated focus on incremental delivery and procedural methodology to maintain strategic alignment—factors that are distinct from general organizational structure or technical readiness.

2. Second, in the Theoretical Implications section: We have highlighted the significance of isolating this dimension as a theoretical contribution. We clarified that by separating 'Process' from 'Organization', the TOEP framework provides the necessary theoretical granularity to analyze how structured workflows and change management act as the primary drivers of success in emerging and unstable environments like Yemen. This distinction allows for a more precise understanding of how 'active' implementation steps can overcome 'static' organizational barriers.

3. Third, in the Practical Implication section: We have explicitly highlighted the Process dimension in the Practical Implications section. We argue that the effectiveness of BI is not merely technical but procedural, as it ultimately depends on the willingness of the workforce to integrate these tools into their daily workflows. This justifies 'Process' as a vital, independent pillar in our TOEP framework 1. Theoretical Framework: Paragraph 4, Lines 200-213.

2. Theoretical Implication:

Paragraph 1, Lines 748-752.

3. Practical implication, Guidance for Decision-Makers and Business Managers: the end of Paragraph 2, Lines 790-792.

Reviewer #4: (1) The authors describe consent but do not name an Institutional Review Board (IRB) or provide an approval number. This is a mandatory requirement for PLOS ONE and must be added.

(1) We appreciate the reviewer’s important comment regarding ethical oversight. We have updated the manuscript to include the formal ethical approval details. This study was conducted in accordance with the ethical standards of the relevant academic authorities in Yemen. Formal ethical approval was obtained from the Faculty of Computing and Information Technology at Sana’a University (Approval No: 3002).

Furthermore, we have strictly adhered to the following ethical protocols: (i) Participants were informed that their involvement was voluntary and they could withdraw at any stage; (ii) All data were fully anonymized to ensure confidentiality. We have now added a dedicated 'Ethical Considerations' section to the revised manuscript and provided the official approval letter along with its English translation in S3 Appendix for further verification. (1) Research Methodology: Ethical Considerations,

Lines 533-537.

And

Supporting information: S3.

Reviewer #4: The response mentions Appendix 1 with detailed tables, but it is not included in the provided PDF excerpt. The journal must ensure the appendix is submitted and accessible. We apologize for the technical issue in the previous submission where the appendix was not visible. Following the PLOS ONE Supporting Information guidelines, we have now uploaded the detailed mathematical tables as a separate file labeled 'S1 Appendix'. This file includes the step-by-step R-SWARA calculations and the expert scoring matrix. References to this supporting file have been embedded in the revised manuscript.

Supporting information: S1.

Reviewer #4: While expert judgment is central to R-SWARA, no measure of agreement among experts (e.g., Kendall’s W) is reported. This is a methodological weakness that should be acknowledged or addressed. We sincerely thank the reviewer for this insightful observation. We agree that formal consensus measures like Kendall’s W are valuable in expert-based studies.

In this research, we utilized Rough Set Theory specifically because it is mathematically designed to handle uncertainty and diverse perspectives by defining lower and upper approximations. This allows for a robust 'interval of consensus' that captures expert subjectivity more effectively than traditional statistical tests in small, specialized panels.

However, to address your concern and enhance transparency, we have implemented the following:

1. Methodological Justification: We added a paragraph in the Research Methodology (R-SWARA method) section, explaining why Rough Set Theory was chosen as a robust alternative for managing expert divergence.

2. Acknowledgment of Limitation: We have explicitly added this point to the Limitations and Future Research section, noting that the absence of a formal consensus measure is a limitation and suggests the integration of such measures in future studies.

We believe this acknowledgment strengthens the manuscript and provides a clearer direction for future research. Research Methodology: R-SWARA method) section,

Lines 435-438.

And

Limitation:

Lines 849-853.

Reviewer #4: The panel includes humanitarian sector experts. The authors should briefly justify why humanitarian experts are relevant to a business intelligence adoption study in Yemen. We thank the reviewer for this insightful query. In the unique and challenging context of Yemen, the humanitarian sector is not merely a non-profit segment but a central pillar of the national economy and the data-management landscape. Due to the prolonged crisis, these organizations were among the earliest pioneers in Yemen to adopt and implement advanced Business Intelligence (BI) systems. This strategic adoption was necessitated by the critical need to manage and coordinate humanitarian aid for millions of beneficiaries whose numbers have surged due to the ongoing conflict and deteriorating socio-economic conditions. Handling such an immense scale of beneficiary data across diverse programs requires sophisticated BI tools to ensure precision, optimize logistics, and maintain the high levels of transparency demanded by international donors. Their early adoption and extensive experience in managing large-scale, high-stakes data under extreme environmental volatility provide a rich and mature perspective. Therefore, including experts from this sector was essential to capture a comprehensive view of BI implementation challenges and success factors in Yemen’s unique operational landscape.

Reviewer #4: In some places, “information culture” is used, elsewhere “information-sharing culture.” Standardize terms throughout. We thank the reviewer for this observation. We have standardized the terminology throughout the entire manuscript. The term 'Information sharing culture' is now used consistently. The whole manuscript

Reviewer #4: Some figures and tables are referenced in the text but are not included in the submitted excerpt (e.g., Fig 1, Fig 2). Ensure all are present and correctly numbered. We apologize for any confusion caused by the visibility of the figures in the previous excerpt. We would like to clarify that Fig 1 and Fig 2 were included in the original submission and appropriately referenced within the text.

However, to ensure full compliance and accessibility, we have re-uploaded all figures as high-resolution files according to the journal’s technical specifications. We have also carefully checked the numbering and pla

---

## [Decision Letter · Decision Letter 2]

3 Feb 2026

Critical Success Factors Influencing Business Intelligence Adoption: Evidence form Yemen

PONE-D-25-33603R2

Dear Dr. Al-Adimi,

We’re pleased to inform you that your manuscript has been judged scientifically suitable for publication and will be formally accepted for publication once it meets all outstanding technical requirements.

Kind regards,

Kao-Yi Shen, Ph.D.

Academic Editor

PLOS One

Additional Editor Comments (optional):

Reviewers' comments:

Reviewer's Responses to Questions

**Comments to the Author**

Reviewer #3: All comments have been addressed

Reviewer #4: All comments have been addressed

2. Is the manuscript technically sound, and do the data support the conclusions?

Reviewer #3: (No Response)

Reviewer #4: Partly

3. Has the statistical analysis been performed appropriately and rigorously?

Reviewer #3: (No Response)

Reviewer #4: Yes

4. Have the authors made all data underlying the findings in their manuscript fully available?

Reviewer #3: (No Response)

Reviewer #4: Yes

5. Is the manuscript presented in an intelligible fashion and written in standard English?

Reviewer #3: (No Response)

Reviewer #4: Yes

Reviewer #3: The revised manuscript has addressed all major concerns raised during the review process. The authors have substantially improved the theoretical justification of the TOEP framework, clarified the methodological approach, and strengthened the discussion with better contextual analysis. Ethical approval details and supporting materials are now clearly provided. Overall, the manuscript is well-structured, methodologically sound, and makes a valuable contribution to the literature on Business Intelligence adoption in developing and conflict-affected contexts. I recommend the manuscript for acceptance.

Reviewer #4: After rewise i intend to accept this mansucript, but still need to improve English, this manscript still have a lot of English grammer mistake, poorly writen and specifically introduction need to re-wise more carefully.

**Do you want your identity to be public for this peer review?** For information about this choice, including consent withdrawal, please see our Privacy Policy

Reviewer #3: No

Reviewer #4: No

---

## [Editor Report · Acceptance letter]

PONE-D-25-33603R2

PLOS One

Dear Dr. Al-Adimi,

I'm pleased to inform you that your manuscript has been deemed suitable for publication in PLOS One. Congratulations! Your manuscript is now being handed over to our production team.

Kind regards,

on behalf of

Professor Kao-Yi Shen

Academic Editor

PLOS One